# Predicting soil thickness on soil mantled hillslopes

Nicholas R. Patton[1], Kathleen A. Lohse [1,2], Sarah E. Godsey[1], Benjamin T. Crosby[1] & Mark S. Seyfried[3]

Soil thickness is a fundamental variable in many earth science disciplines due to its critical role in many hydrological and ecological processes, but it is difficult to predict. Here we show a strong linear relationship ($r^2 = 0.87$, RMSE = 0.19 m) between soil thickness and hillslope curvature across both convergent and divergent parts of the landscape at a field site in Idaho. We find similar linear relationships across diverse landscapes ($n = 6$) with the slopes of these relationships varying as a function of the standard deviation in catchment curvatures. This soil thickness-curvature approach is significantly more efficient and just as accurate as kriging-based methods, but requires only high-resolution elevation data and as few as one soil profile. Efficiently attained, spatially continuous soil thickness datasets enable improved models for soil carbon, hydrology, weathering, and landscape evolution.

[1] Department of Geosciences, Idaho State University, Pocatello, ID 83209, USA. [2] Department of Biological Sciences, Idaho State University, Pocatello, ID 83209, USA. [3] Agricultural Research Service, Northwest Watershed Research Center, 800 Park Blvd., Plaza IV, Suite 105, Boise, Idaho 83712, USA. Correspondence and requests for materials should be addressed to K.A.L. (email: klohse@isu.edu)

Soil thickness results from the balance between the rates of soil production and transport[1–3] and is a critical attribute of many hydrological and ecological processes[4]. In geomorphology, it is crucial in determining hillslope stability, drainage density, and channel initiation[5–8]. In the hydrological and ecological sciences, it governs runoff responses[9], water residence, and travel time distributions[10–12], it determines plant-available water, storage, and sourcing[13,14], and it often sets the lower boundary conditions for soil carbon and other elemental accumulation and storage[15]. Despite its importance, spatially distributed soil thickness data are rarely obtained due to the physical and monetary cost of excavating soil to the required depths. To date, soil thickness cannot be efficiently predicted across a landscape. As such, it remains a poorly constrained yet key parameter that hinders advancements in landscape evolution, hydrological, and earth system models[12,16,17].

Soil thickness is heterogeneous in space and time across landscapes, and models often assume that thickness scales inversely with erosion rate[2] or relief[18,19]. Attempts to quantify and model the relationship between soil thickness, production, and topography have typically used hillslope mass balance models and cosmogenic radionuclide (CRN)-derived erosion rates[2,3,20–22]. More recently, numerical models for geochemical mass balance[23], soil production[12,24–27], landscape-scale pedogenesis[28], and soil transport[29,30] have been applied to this problem.

Assuming sediment flux is linearly proportional to slope, conservation laws predict an inverse linear relation between the soil production rate and hillslope curvature[2,5,21] ($C$), quantified as the rate of change in slope from a fixed point on a landscape in all directions. Researchers have also independently demonstrated that soil production rates decrease exponentially with increasing soil thickness[2,3,20–22,29,30]. Combining these two relationships establishes soil thickness as proportional to the natural logarithm of curvature[21,26], explicitly on convex, divergent hillslopes (associated with negative curvatures). In contrast to convex areas, concave, convergent areas have received less attention in part because soil production is reduced under thick soil cover and theoretical predictions using the natural logarithm of curvature fail for positive values.

Recent, field-calibrated numerical models that estimate soil thickness show thick soils in concave hollows and thin soils on convex ridges and predict relative thicknesses in catchments of varying lithology reasonably well[26,27]. However, accurate predictions of absolute thicknesses have not been obtained[27], and soil thickness models remain over-parameterized and require extensive and computationally expensive analyses[24,25,27].

Here we explore the empirical relationship between curvature derived from high-resolution digital elevation models (DEMs) and field-measured thickness of mobile regolith (TMR) and test the assertion that soil thickness varies as the natural logarithm of curvature. Because the terms soil thickness and soil depth are not used consistently across disciplines, we use the term TMR to define the portion of the soil profile that is mobile via slope or mixing processes[31–33] (Supplementary Fig. 1). We show a strong linear relationship between soil thickness and hillslope curvature across both convergent and divergent topography at our site in Idaho and then observe similar relationships across other catchments, although the slopes and y-intercepts vary widely. We demonstrate that the slopes of these functions vary with the standard deviations in catchment curvatures and that the catchment curvature distributions are centered on zero. The significance of the curvature distributions being normally distributed and centered on $0\,\mathrm{m}^{-1}$ is that the intercept of the curvature-TMR function represents the mean TMR within each catchment. We present and validate a simple empirical model for predicting the spatial distribution of soil thickness in a variety of

catchments based only on high-resolution elevation data and few soil profiles. Our findings indicate that our linear TMR-curvature model produces TMR estimates that are just as reliable as kriging-based interpolations with significantly less labor and cost. Our model also provides more robust estimates of TMR across the full range of curvature values than using a natural logarithm relation.

## Results

**Sampling thickness of mobile regolith and curvature**. We measured curvature and TMR in a small (1.8 km²), semi-arid granitic catchment, Johnston Draw, in the Reynolds Creek Critical Zone Observatory (CZO) and Experimental Watershed (RCEW) in Idaho, USA. We sampled thirty-nine locations across the full range of curvature values and elevations (Supplementary Fig. 2) and determined TMR by digging soil pits vertically from the surface to the contact between mobile regolith and immobile weathered bedrock (Supplementary Fig. 1). This contact was determined from observation of original parent material structure including exfoliation sheets, planar flow fabrics, or jointing sets[31].

We compared this dataset to equivalent datasets collected from catchments representing a wide range of climates and vegetation types on predominantly felsic parent materials[2,3,20–22] and then cross-validated on catchments with felsic[34,35] as well as mixed mafic-felsic lithologies (Table 1). Thickness determinations were standardized to the highest degree possible (See Methods, Supplementary Fig. 3), and uncertainties were propagated through all analyses.

Curvature was calculated as the rate of change in slope from a fixed point relative to eight neighboring cells[36,37] using a geographical information system (ArcGIS v.10.2.2, ESRI, Redlands, CA). We utilized ArcGIS primary curvature output, which is derived from Zevenberger and Thorne[38] and Moore et al.[37] equations. The ArcGIS curvature function differentiates the slope in percent rather than the actual gradient, and reverses the sign so to compute curvature values in units $1\,\mathrm{m}^{-1}$, we divide the ArcGIS output by $-100$. We extracted curvature values for Johnson Draw from a Light Detection and Ranging (LiDAR) DEM resampled to 3-m resolution because a sensitivity analysis showed that this resolution provided the highest correlation between TMR and curvature (Table 2). When comparing Johnson Draw with other datasets, we resampled the LiDAR data to 5 m resolution because some of the cross-site datasets were manually collected at this resolution[2,3,20–22].

**Linear TMR-curvature relationship**. Similar to other studies (Table 1), we observed the thinnest TMR on ridges and noses and the thickest TMR in hollows and valleys in Johnston Draw. In contrast to previous findings, we found that TMR varied as a robust, positive linear function of curvature across both convergent and divergent parts of the landscape ($N = 26$, $r^2 = 0.87$, RMSE = 0.19 m, Fig. 1a). The slope of the TMR-curvature relationship was 22.8 m² with a y-intercept of 1.01 m. Note that for this relationship, the y-intercept is defined as the value when $x = 0$; in this case, the y-intercept is the TMR on a planar surface where the curvature is $0\,\mathrm{m}^{-1}$. We tested the predictive power of this relationship by comparing predicted and observed values (30% of Johnston Draw dataset was reserved for validation, $N = 12$, Fig. 1b). The null hypothesis that the predicted vs. observed slope was significantly different than 1 was rejected ($t = -0.53 <$ critical $t_{0.05,11} = 2.20$). We also evaluated model selection within the convex, negative curvature regions and found more support for TMR varying linearly ($N = 22$, $r^2 = 0.63$, RMSE = 0.18) rather than in proportion to the natural logarithm of curvature ($r^2 = 0.37$, RMSE = 0.22) (Supplementary Fig. 4a). This is in conflict with the current theoretical paradigm and may result from failed

**Table 1 Site characteristics**

| Site | Major aspect | Mean elevation (m) | MAP (mm/y) | MAT (°C) | Lithology | Mean curvature (m$^{-1}$) | Curvature standard deviation (m$^{-1}$) | Reference |
|---|---|---|---|---|---|---|---|---|
| Johnston Draw, ID, USA | North and South | 1600 | 550 | 7.4 | Granite-Diorite and Quartz-Monzonite | 0.0001 | 0.021 | This study |
| Tennessee Valley, CA, USA | Northeast and Southwest | 170 | 760 | 14 | Greenstone, Greywacke and Chert | −0.00012 | 0.034 | [2,3] |
| Coos Bay, OR, USA | West and East | 669 | 1500 | 11.5 | Sandstone and Siltstone | −0.00013 | 0.059 | [21] |
| Nunnock River, NSW, AU | Northwest and Southeast | 950 | 910 | 13.5 | Granite-Diorite | ND | ND | [20] |
| Point Reyes, CA, USA | South | 112 | 430 | 11.5 | Quartz-Diorite and Granite-Diorite | 0.00004 | 0.039 | [22] |
| Marshall Gulch (sub-catchment), AZ, USA | North and South | 2439 | 875 | 10 | Granite, Quartzite and Amphibolite | −0.00009 | 0.071 | [27] |
| Babbington Creek, ID, USA | North and South | 1500 | 513 | 7.4 | Granite-Diorite and Quartz-Monzonite | −0.00004 | 0.018 | This study |
| Gordon Gulch, CO, USA | North and South | 2583 | 519 | 5.1 | Biotite-Gneiss | 0.00006 | 0.040 | [34,35] |
| Reynolds Mountain, ID, USA | West and East | 2082 | 866 | 5.2 | Rhyolitic Tuff and Basalt | 0.00002 | 0.020 | This study |

These include major aspect, mean elevation, mean annual precipitation (MAP), mean annual temperature (MAT), lithology, and catchment curvature mean and standard deviation derived from high resolution (5 meter) digital elevation model (DEM) for cross-site analysis and model validation. Light and Ranging (LiDAR) data was not available for the Nunnock River. Mean elevation and curvatures were derived from reported local observations[20], and mean and distribution of curvature were not obtained

**Table 2 Sensitivity of curvature-thickness of mobile regolith (TMR) relationship to resolution of digital elevation model (DEM) for the Johnston Draw data set ($N = 38$)**

| DEM resolution (m$^2$) | Catchment curvature standard deviation (m$^{-1}$) | TMR uncertainty (m) | Curvature uncertainty (m$^{-1}$) | Slope (m$^2$) | Intercept (m) | RMSE (m) | $r^2$ | p-Value |
|---|---|---|---|---|---|---|---|---|
| 1 | 0.126 | 0.13 | 0.1521 | 0.41 | 1.05 | 0.54 | 0.02 | <0.0001 |
| 3 | 0.036 | 0.13 | 0.0169 | 22.80 | 1.01 | 0.20 | 0.86 | <0.0001 |
| 5 | 0.021 | 0.13 | 0.0061 | 21.58 | 1.04 | 0.40 | 0.44 | <0.0001 |
| 10 | 0.018 | 0.13 | 0.0015 | 20.56 | 1.00 | 0.45 | 0.30 | 0.0004 |
| 20 | 0.011 | 0.13 | 0.0004 | 24.76 | 0.98 | 0.48 | 0.21 | 0.0039 |
| 30 | 0.007 | 0.13 | 0.0002 | 37.90 | 0.96 | 0.47 | 0.25 | 0.0013 |
| 50 | 0.006 | 0.13 | 0.0001 | 59.73 | 0.95 | 0.47 | 0.23 | 0.0022 |

TMR uncertainty reported as standard error is based on propagation of error of the average observed TMR (see Methods). Horizontal and vertical uncertainty in DEM were obtained through metadata associated with 2007 Light and Ranging (LiDAR) dataset. Curvature uncertainty as measured by standard error was calculated by the Method of Moments assuming correlation between uncertainty of neighbor and center cell points are 0 ($r = 0$). Slope of curvature-TMR, intercept value, root-mean-squared error (RMSE), coefficient of determination ($r^2$), and p-value based on linear regression

assumptions surrounding steady state soil thickness, the linear relation between slope and sediment flux and/or the exponential formulation of the soil production model. The linear TMR-curvature relation is also advantageous compared to the natural logarithm because it can handle both positive and negative values of curvature.

We anticipated that TMR-curvature relations would exhibit greater variation at larger positive curvatures (concave, convergent areas) owing to either over-thickening or recently failed soils in zero-order hollows. However, there was no change in variability with curvature at our site, evidenced by homogeneity in variance around the best-fit line (Fig. 1a). These findings suggest that the soil thickness in concave hollows of Johnston Draw are regulated by frequent and efficient transport processes instead of erratic evacuations as suggested by Dietrich et al.[5]. Instead of thickening indefinitely, hollows may be maintained by creep or surface erosion. Similarly, TMR thicknesses on concave toe slopes on terraces or floodplains may be regulated by creep or surface erosion rather than evacuation by lateral channel migration. This supposition is consistent with findings by Dietrich et al.[5] who found predicted soil thicknesses >1 m on toe slopes, but warrants further study in this and other landscapes to understand the processes underlying this linear relationship.

**Cross-site comparison.** Consistent with our observations at Johnston Draw, we found similar linear relationships between TMR and curvature in the cross-site dataset (Fig. 2a), but the best-fit slopes and intercepts of those relationships varied from site to site. The y-intercepts for the cross-site dataset (the soil thickness on planar surfaces) ranged from 0.57 m to 1.11 m. In the convex regions of the coherent sites (Table 1), linear models again outperformed natural logarithm models (Supplementary Fig. 4b-g).

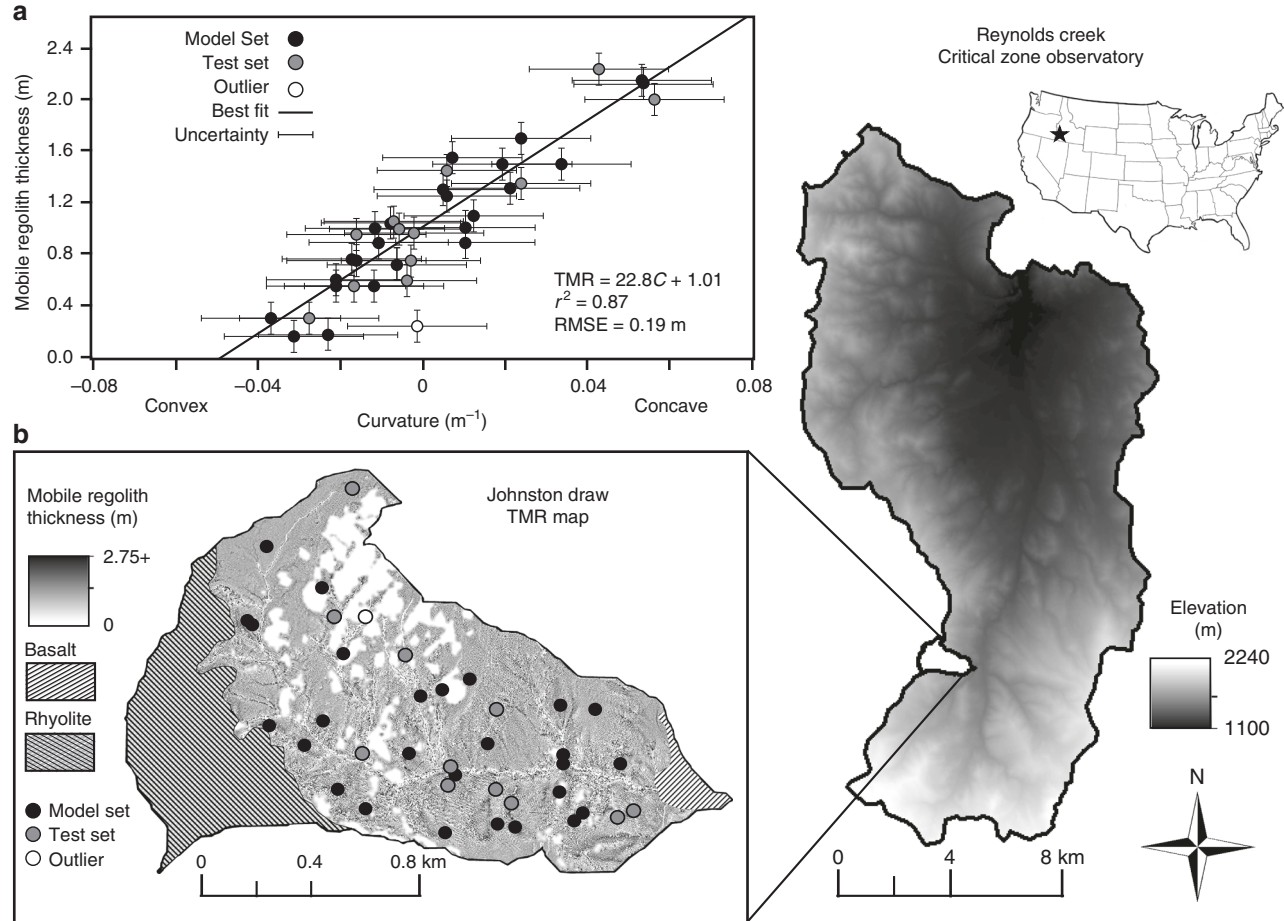

**Fig. 1** Curvature and thickness of the mobile regolith plot and predictive map. **a** The thickness of the mobile regolith (TMR) varies as a strong linear function of curvature ($C$) in Johnston Draw. Black dots represent randomly selected build dataset (70% of sites). Gray dots represent test set to validate the model. The white dot is a location that was excluded owing to proximity to both a rock outcrop and a stream channel. Uncertainty is reported as the standard error by the Method of Moments. **b** Predicted TMR map for the granitic portion of Johnston Draw derived from the TMR-curvature function using a 3 m Light Detection and Ranging (LiDAR)-derived digital elevation model. Darker shades indicate larger TMR (2.75 + m) and lighter shades indicate smaller TMR (0 m) including those areas excluded as rock outcrops or streams. Hatched areas indicate non-granitic portions of the watershed that were not modeled

In Johnston Draw, we observed that curvatures were normally distributed within the catchment (Fig. 2b) with a mean of 0 m$^{-1}$ and a standard deviation of 0.0209 m$^{-1}$. Similarly, for the cross-site dataset, catchment curvature distributions were also normally distributed with a mean of 0 m$^{-1}$, indicating a tendency of the curvature distributions to center around planar surfaces (Fig. 2b). Therefore, using the linear TMR-curvature relationship, we find that the y-intercept defines the mean soil thickness within each catchment.

In contrast to the observation that curvature distributions were normal and centered on 0 m$^{-1}$ for all catchments, surface roughness, defined here as the standard deviation in catchment curvature ($\sigma_c$) at a given scale[39], varied from 0.0209 to 0.0713 m$^{-1}$ across sites (Table 1, Fig. 2c). Sites varied in the slope of their TMR-curvature function from the steepest best-fit slope at Johnston Draw to the shallowest at Marshall Gulch. Among the sites, the slope of the TMR-curvature function varied linearly with $\sigma_c$ (Fig. 2c). Sites with high $\sigma_c$ (e.g. Marshall Gulch and Coos Bay) had TMR-curvature slopes near zero indicating that curvature poorly predicted TMR. In these sites, large magnitude, high frequency stochastic disturbances (e.g. tree-throw, mass movements and burrowing) alter both the surface topography and the TMR[21,27]. In contrast, sites with low $\sigma_c$ (e.g. Johnston

Draw and Point Reyes) had both high TMR-curvature slopes and $r^2$ values. In these catchments, soil formation and transport processes are likely driven by low-magnitude, gradual processes (rheologic creep, lesser bioturbation) that smooth surface topography, resulting in curvature values that explain much of the variation in TMR (Fig. 2a).

Interestingly, comparison of $\sigma_c$ and published soil production rates showed that catchments with high $\sigma_c$ such as Coos Bay also had the highest soil production and erosion rates[21]. In contrast, those with lower $\sigma_c$ such as Tennessee Valley had lower production and erosion rates[2,3], possibly indicating that catchment curvature distributions may also provide a useful proxy for soil production rates[40,41]. Consistent with this idea, one study showed that catchment roughness as measured by the standardized topographic position index can be used as a proxy for sediment availability[42] suggesting that catchment surface roughness may have some utility as a proxy for mapping different geomorphic processes or process rates. Recent work extends theoretical linkages supporting the idea that landscapes with narrow slope distributions are dominated by diffusion-like processes whereas rough, juvenile landscapes with broad slope distributions are affected by a stochastic combination of diffusive soil creep, advective river incision, and noise[43]. The

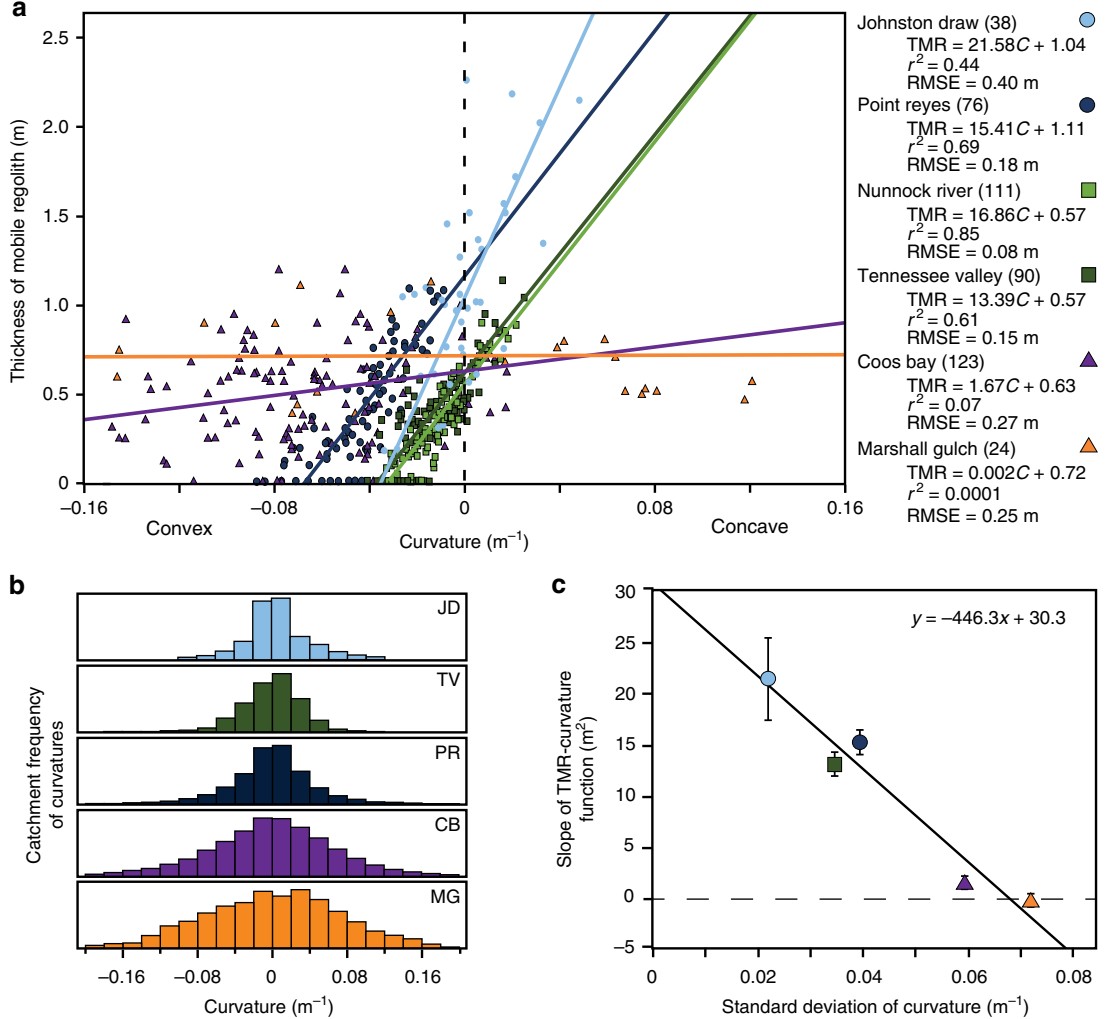

**Fig. 2** Cross-site evaluation. **a** Cross-site evaluation of six catchments in which the thickness of the mobile regolith (TMR)-curvature (C) function is evaluated using a 5-m digital elevation model (DEM). **b** Depicts catchment curvature distributions based on a 5 m DEM centered on 0 m⁻¹. **c** Cross-site comparison of the slope of the TMR-curvature function (and associated standard error) with the local standard deviation in catchment curvature ($\sigma_c$). Nunnock River (light green squares) dataset was not included in plots b or c due to the lack of high resolution Light Detection and Ranging (LiDAR) data; curvature estimates for Nunnock River in **a** were derived from reported local observations[20]. Note, the curvature distributions are derived from all cells within the catchment's DEM

linkages between the statistical properties of topography (slope and curvature) and geomorphic processes merit continued investigation.

**Simple empirical model to predict TMR.** Based on the above analyses, we present a simple empirical model to predict TMR at any location within a catchment using high-resolution LiDAR data and a limited number of TMR measurements. We start with the following equation that generalizes the relationship between TMR at a point ($h$) and the curvature at that point ($C$):

$$h = \left(\frac{\Delta h}{\Delta C}\right) C + \bar{h} \qquad (1)$$

where $\frac{\Delta h}{\Delta C}$ is the slope of the TMR-curvature relationship illustrated in Fig. 1a and $\bar{h}$ is the average $h$ found within a catchment. The slope parameter $\frac{\Delta h}{\Delta C}$ can be estimated directly from the equation in Fig. 2c based on catchment $\sigma_c$. Because all catchments examined have a normally distributed curvature centered at 0 m⁻¹ (Fig. 2b), $\bar{h}$ can be determined by measuring TMR at

selected locations with planar ($C = 0$ m⁻¹) surfaces (at least one estimate of $\bar{h}$ is required, additional pits will improve estimate and constrain uncertainty). We note that we adapted self-correlation diagnostic methods described by Worrall et al.[44] to assess this relationship, and we could not entirely reject the possibility that the strength of the relationship in Fig. 2c might be partially due to self-correlation because the linear regression slopes of the TMR-curvature relationships in Fig. 2a partially depend on the variability of the sample population of catchment curvatures. However, given the benefits of predicting TMR from easily derived topographic characteristics, this approach merited cross-validation with data from other locations.

**Independent tests of model.** As independent tests of this model, we predicted TMR using Eq. (1) from topographic and TMR data collected from three catchments, two of which had low $\sigma_c$, Babbington Creek and Reynolds Mountain (RM), both sub-catchments of the Reynolds Creek CZO, with granitic and mixed basalt and rhyolite lithology, respectively. The third catchment, Gordon Gulch, a gneissic catchment in Boulder Creek

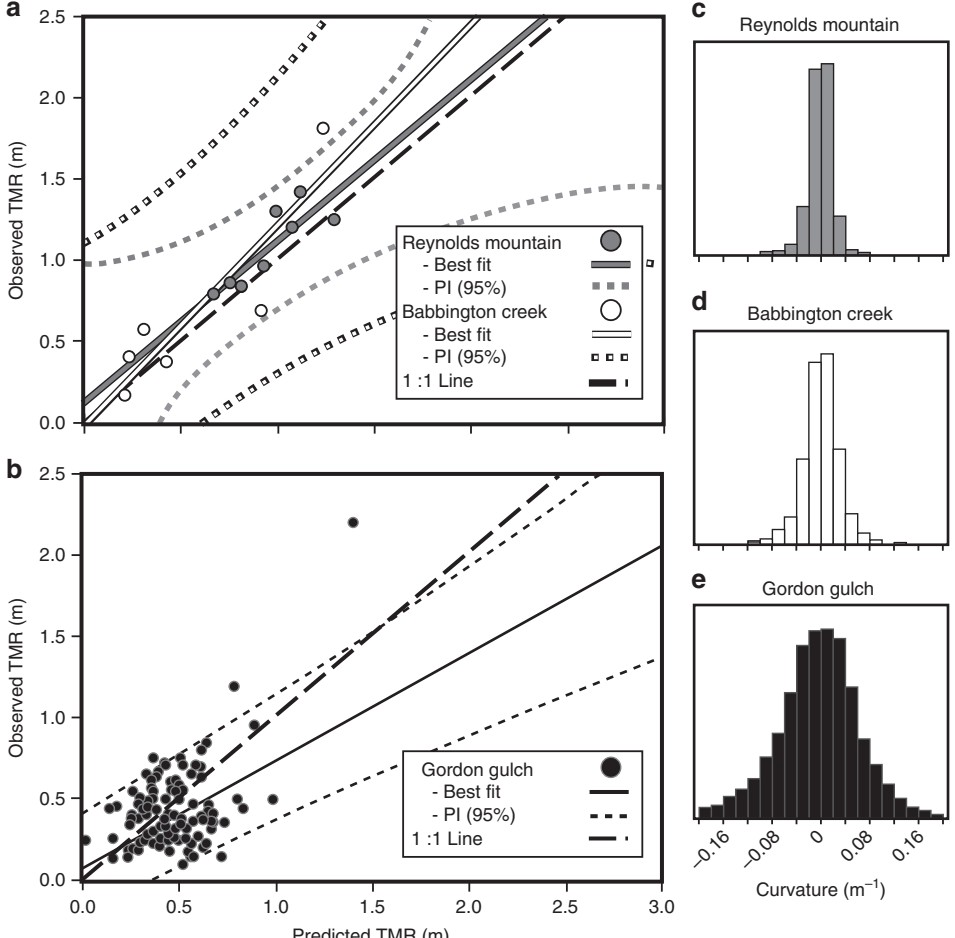

**Fig. 3** Model validation. Validation of the thickness of the mobile regolith (TMR)-curvature (C) approach at Babbington Creek and Reynolds Mountain, Idaho (**a**) and Gordon Gulch, Colorado, USA (**b**). Solid white, gray, and black lines represent best-fit predicted vs. observed TMR values based on curvature calculated from a Light Detection and Ranging (LiDAR)-derived digital elevation model (DEM) and a single soil pit for Reynolds Mountain (**c**), Babbington Creek (**d**), and Gordon Gulch (**e**), respectively. Large dashed black lines represents the 1:1 predicted versus observed line. The best-fit slopes were not significantly different from one for Babbington Creek ($|t| = 0.74 <$ critical $t_{0.05,5} = 2.571$) and Reynolds Mountain ($|t| = 0.01 <$ critical $t_{0.05,7} = 2.365$), indicating unbiased models, whereas the slope was significantly lower than one for Gordon ($p < 0.001$, $|t| = 3.64 <$ critical $t_{0.05,162} = 1.974$), indicating over-prediction with higher TMR than observed. Small white, gray, and black dotted lines represent the 95% prediction intervals (PI)

CZO, had a higher $\sigma_c$[34,35]. We expected that curvature would predict TMR well in Babbington and Reynolds Mountain with lower $\sigma_c$ whereas curvature would explain less variation in TMR in Gordon Gulch with higher $\sigma_c$.

One might assume that TMR in Babbington and Johnston Draw (Fig. 1) would be similar due to their similar lithology and climate. However, $\bar{h}$ was 1.04 m (with 5 m DEM) for Johnston Draw compared to 0.56 m for Babbington based on a single soil pit on a planar surface, and $\sigma_c$ was 0.0209 m$^{-1}$ for Johnston Draw compared to 0.0184 m$^{-1}$ for Babbington; $\bar{h}$ was 0.93 m and $\sigma_c$ was 0.0191 m$^{-1}$ for Reynolds Mountain, the mixed lithology catchment. Based on these inputs to the model, predicted TMR values at Babbington agreed well with observed values from the validation data set ($N = 6$, slope = 1.24, $r^2 = 0.79$, RMSE = 0.30 m, $p = 0.0181$) (Fig. 3a). Predicted TMR values also strongly agreed with observed values from a validation data set derived from mixed mafic and extrusive felsic parent material at Reynolds Mountain ($N = 8$, slope = 1.002, $r^2 = 0.72$, RMSE = 0.14 m, $p = 0.0080$) (Fig. 3a) suggesting that curvature is an excellent proxy for predicting TMR in catchments that display low $\sigma_c$. Findings from Reynolds Mountain also indicate that the model may have general utility across lithologies.

As expected, the TMR function from Eq. (1) could not explain as much variability in Gordon Gulch TMR that had a higher $\sigma_c$ of 0.040 m$^{-1}$ ($N = 163$, slope = 0.64, $r^2 = 0.21$, RMSE = 0.20 m, $p < 0.0001$) (Fig. 3b). It is also worth noting that Gordon Gulch also had a larger vertical uncertainty in the LiDAR dataset, 0.175 m compared to 0.034 m for both Babbington and Reynolds Mountain, which may help to explain the lower model performance. Despite this poorer relative performance, one of the benefits of the TMR-curvature model for catchments with higher $\sigma_c$ is that it can improve site selection and thus reduce physical and monetary costs of interpolated TMR estimates (based on the $\sigma_c$). Perhaps more importantly, the TMR-curvature model, based on a 5 m resolution LiDAR to estimate $\sigma_c$ and one soil profile on a planar surface to estimate the intercept, performs just as well as kriging-based interpolations. Indeed, comparison of our approach to simple, ordinary, and regression kriging models at Gordon Gulch shows that kriging models do not improve TMR estimates compared to our TMR-curvature model ($r^2 = 0.19$ and RMSE of ~0.44 m compared to $r^2 < 0.02$ and RMSE of ~0.4 m for both simple and ordinary kriging and $r^2 = 0.06$ and RMSE of ~0.37 m for regression kriging). We only conducted this comparison at Gordon Gulch where there were sufficient

samples to compare techniques[45] (113 build and 50 test pits, 163 total).

## Discussion

Findings from our study indicate that our linear TMR-curvature model may produce TMR estimates that are just as reliable as kriging-based interpolations with significantly less labor and cost. Our model also provides more robust estimates of TMR across the full range of curvature values than using a natural logarithm relation. Indeed, our analyses showed more support for TMR varying linearly rather than in proportion to the natural logarithm of curvature (Supplementary Fig. 4) raising questions about the underlying assumptions of steady state soil thickness, the linear relation between slope and sediment flux, and/or the exponential formulation of the soil production model. Our finding that catchment curvatures are normally distributed around planar surfaces has high utility in numerous earth science disciplines because a first-order estimate of the mean TMR ($\bar{h}$) for a given catchment can be derived from a single soil profile at a planar position. These results together suggest that the linear TMR-curvature model may be a good first order estimator of TMR where good quality, fine resolution DEM data exist and limited resources are available for digging many soil pits.

Findings from our study also indicate that surface roughness as measured by the standard deviation in catchment curvature at a given scale reflects the degree to which local topography governs soil thickness in a given catchment. In our Idaho study sites (Johnston Draw, Babbington Creek, and Reynolds Mountain), local topography as measured by curvature is the primary determinant of TMR. In contrast, topography explains less of the variation in TMR in catchments with broad curvature distributions. Indeed, uncertainty in the model increases as $\sigma_c$ increases such that the predictive capability of the model declines in these regions; other physical or biological model parameters may be needed to explain the variability in TMR where surface roughness is high.

We posit that soil thicknesses in catchments with high surface roughness are not governed solely by local topography but rather by multiple geomorphic processes (e.g. mass movements, tree throw, etc.) and in-situ soil evolution that may influence soil production rates[33]. Marshall Gulch, Coos Bay, and Gordon Gulch appear to be good examples of this where TMR is relatively insensitive to curvature as indicated by the slope of TMR-curvature function approaching zero (Fig. 2). In these catchments, steep hillslopes and the abundance of trees can result in more frequent and spatially heterogeneous topographic disturbances such as landslides and tree throw that increase surface roughness and therefore influence the variability in TMR. In Johnston Draw, Marshall Gulch and Gordon Gulch, rock outcrops create similar local disturbances, features not captured in our model or others[26,27]. As a consequence, pit locations within 10 m of rock outcrops were excluded from our analysis. Other possible limitations to the model include uncertainty in estimating TMR in complex environments: in colluvium where defining the basal boundary is difficult, in floodplains where disturbance introduces and exhumes materials, in aeolian landscapes where soil development is driven by atmospheric inputs[25], and in glaciated landscapes where topography results from erosion or deposition by ice. Finally, our model has not been tested in more stable landscapes with deeper soils, which will likely exhibit thicker immobile regolith, another feature not captured by our mobile regolith model. We would not expect our model to work for predicting total regolith (mobile + immobile) because variations in climate and substrate will control saprolite thickness.

Further testing and appraisal of the limitations of this model await the collection of additional data that combine spatially distributed, high-resolution elevation data with extensive TMR measurements. Currently, the low availability of TMR measurements limits the application of this model. Studies that dig to refusal, bedrock or at a predetermined depth may include or exclude portions of the mobile regolith resulting in inconsistencies. In addition, methods such as augering, soil tile pole, and knocking pole may underestimate thickness in rocky soils and/or overestimate depth if penetrating fractures in bedrock. Without properly identifying the mobile regolith boundary with soil pits, the likelihood of producing topographic relationships is low. For this reason, comparisons of our work to other studies that used these other methods[46,47] are problematic and require considerable knowledge of both the research methods and site location.

In contrast to TMR measurements, high-resolution elevation data are becoming increasingly abundant and technology associated with unmanned aerial vehicle, airborne and remotely sensed data collection is advancing rapidly and will likely provide sufficient elevation data on demand. Our sensitivity analysis at Johnston Draw indicates that the TMR-curvature relationship is highly sensitive to scale (Table 2), with some deterioration of the relationship with the 5 m resolution resampling and considerable deterioration at 30 m resolution, which is typical of widely available DEM's. Our cross-site comparison using consistent TMR measurements and high resolution LiDAR data may explain why our findings have not been previously reported.

The spatial distribution of soil thickness at the landscape scale remains largely unknown due to the difficulties in its direct measurement. Beyond practical applications to forestry, construction and transportation industries where soil thickness is a key parameter for predicting landslides, we contend that our TMR model has the potential to advance many earth science disciplines by efficiently providing accurate and spatially distributed thickness estimates, especially in those catchments with low curvature variability. With high-resolution elevation data and a limited set of soil thickness measurements, this model can constrain thickness parameters, which are currently key uncertainties in surface process models. As the number of sites with TMR measurements increase, our model will likely be complementary to the digital and global soil mapping community[48], enabling rigorous testing of global soil thickness models[26]. In hydrologic models, spatially distributed thickness estimates can help to constrain rooting depths[49] and define catchment control volumes and thus reduce uncertainty in streamflow, water storage, water and nutrient residence time, and sourcing to streams[9–14]. This model can be also coupled with measurements of carbon to obtain rapid and cost efficient budgets of the total soil carbon reservoir across complex terrain[50]. These advances will improve soil carbon budgets and intermediate scale carbon cycling and earth system modeling[17,51,52].

## Methods

**Study area**. The study was conducted at the Reynolds Creek Critical Zone Observatory (RC CZO) co-located within the USDA Agricultural Research Service (USDA-ARS) Reynolds Creek Experimental Watershed, a 239 km² catchment located in southwestern Idaho, USA. We focused our study on Johnston Draw, a 1.8 km² sub-catchment oriented east to west (Fig. 1). The bedrock in the sub-catchment is primarily ~66-62 Ma biotite muscovite granodiorite and quartz monzonite from the Idaho Batholith[53]. The relatively spatially continuous lithology weathers to a consistent sandy loam soil texture (average: 67% sand, 18% silt, and 15% clay). Minor lithologic discontinuities in the sub-catchment include a combination of quartz latite, and rhyolite flows covering the high plateau at the top of the catchment and a small ~15.2 Ma olivine-rich basalt flow near the outlet[54] (Fig. 1b); these sparse lithologies were excluded from this analysis to control for lithology while varying topography.

Johnston Draw has asymmetric aspects; steep north-facing slopes average 16.8° and cover 37% of the catchment area and shallow south-facing slopes with average slopes of 13.9° cover the rest of the catchment. The south-facing slopes have larger and more frequent rock outcrops compared with the north-facing slopes. The lack of geomorphic evidence of landslides supports our assertion that hillslope transport in Johnston Draw is dominated by soil creep. Based on our field observations, stream channels with drainage areas >~6000 m$^2$ are incised to bedrock, and all sediment delivered to these channels is transported out of the system, primarily via snowmelt runoff.

The mean annual precipitation is 550 mm yr$^{-1}$ and the mean annual temperature is 7.4 °C. Johnston Draw is located within the rain-snow transition zone[55,56] with elevations ranging from 1490 m to 1850 m. Precipitation primarily occurs in the fall and winter. Lower elevations receive precipitation as rain while higher elevations receive precipitation as snow, and drifts typically accumulate in select locations in the upper basin. Wyoming Big Sagebrush (*Artemisia tridentata* ssp. *wyomingensis)* is the dominant plant species on both aspects representing 50–75% of the catchment, with mountain mahogany (*Cercocarpus ledifolius*), aspen (*Populus tremuloides*), bitterbrush (*Purshia stansburyana*), and western juniper (*Juniperus occidentalis*) making up the remaining major vegetation[55].

**Study design and sampling**. We determined the curvature frequency distribution within Johnston Draw using a 3 m posting DEM derived from a 2007 Light Detection and Ranging (LiDAR) dataset using the ArcGIS curvature toolbox. A 3 m DEM was selected based on a sensitivity analysis of the TMR-curvature function to the resolution (scale) of the DEM (Table 2); the original 1 m DEM was resampled to 3, 5, 10, 20, 30, and 50 m DEM utilizing the mean elevation of the nine adjacent cells. The TMR-curvature relationship was strongest for curvature derived from the 3 m DEM ($N = 38$, $r^2 = 0.86$, RMSE = 0.20 m) whereas it deteriorated some with the 5 m resolution resampling ($N = 38$, $r^2 = 0.44$, RMSE = 0.40 m) and considerably at a resolution of 30 m ($N = 38$, $r^2 = 0.25$, RMSE = 0.47 m). For local estimates of soil thickness (Fig. 1b), the 3 m DEM was used.

Sample sites for TMR were selected from a dataset of soil pits that were originally identified to map watershed soil carbon across an elevation gradient. The pits were chosen via stratified random sampling across six elevation strata, and represent the full range of curvature (Supplementary Fig. 2). TMR was determined by digging 39 soil pits (note that a handful of pits were located on non-granitic parent material and were excluded for this study). Pits were ~1 m deep by 2 m long throughout the granitic portion of the catchment. Similar to Heimsath et al.[3], measurements were taken vertically from the top of the profile to the contact of mobile regolith and weathered bedrock to validate thickness measurements (slope normal to soil thickness).

The largest sources of variability in TMR determination were the designations of the upper and lower boundaries for thickness measurements. Due to matted vegetation, surface cracking, bioturbation and other processes, a fine scale surface roughness of ±0.05 m was observed at most soil pit surfaces, making the location of the upper boundary uncertain. For consistency, all measurements were made at the average surface height as exemplified in Supplementary Fig. 3a. Similar to surface measurements, lower boundary measurements were often difficult. In shallow (<0.50 m) mobile regolith, sharp boundaries were observed with variability around ±0.05 m (Supplementary Fig. 3b). However, as the TMR increased, the uncertainty in the lower boundary increased; soil pits deeper than 1.50 m exhibited a diffuse lower boundary with variability up to ±0.20 m (Supplementary Fig. 3c). The observed variability in the upper and lower boundary conditions were incorporated into error propagation of the TMR-curvature model.

**Data analysis**. Seventy percent (70%) of the TMR measurements from Johnston Draw were randomly selected to quantify the relationship between curvature and TMR. We used a linear regression model with curvature and TMR. We propagated vertical LiDAR uncertainty based on reported flight metadata for each cell according to the equations from Moore et al.[37] and Zevenbergen et al.[38] cited by Arc's curvature algorithm. To be consistent with current literature, we removed the negative curvature and percent convention, as used by ArcGIS, by dividing curvature values by −100. We used the Method of Moments approach to determine standard error, allowing the correlation in uncertainty between elevations of adjacent cells to vary between −1 and +1. Because the degree of correlation was unknown, we reported the uncorrelated errors, which were the central tendency of the range of correlated errors. The remaining 30% of the TMR samples were used as a validation test set to evaluate the goodness of fit of the model. Predicted TMRs and their confidence intervals (95%) were compared to the observed TMR. We tested whether predicted vs. observed slopes were significantly different than 1. We adopted a significance level $α = 0.05$ for all statistical tests. Residuals were examined for normality and structure as well as spatial dependence.

A map of TMR for every 3 m pixel within the catchment was estimated using the derived curvature-TMR function within ArcGIS version 10.2.2 (ESRI, Redlands, CA, Fig. 1b). TMR within two areas in the catchment - near rock outcrops and in stream channels - were post-processed and reclassified. Stream channels were reclassified with TMR values of 0 m based on field interpretation of exposed bedrock beds (accumulations of ≥6000 m$^2$). National Agricultural Imagery Program (NAIP) imagery was used to delineate all outcrops, and these cells were reclassified to thicknesses of 0 m. Within 10 m of rock outcrops, local TMR values

were observed to differ from TMR patterns throughout the sub-catchment. Within that buffer area of 10 m, TMR values were linearly interpolated from the outcrop base (0 m) to the predicted TMR value at 10 m. Masking of these areas was not utilized to avoid influence on the curvature distributions and derived relationships.

**Cross-site evaluation**. We compiled the TMR values in Table 1 using Data Thief III[57] and curvature data from OpenTopography and evaluated the generality of the TMR-curvature function across sites. These sites varied dramatically in bedrock age, substrate, climate, topography, and biota (Table 1), but observations of curvature and soil thickness were consistent across all sites (see references for more details on the sites). We resampled LiDAR for all data sets (including Johnston Draw) to a 5-m resolution DEM due to the lack of higher resolution in all datasets. A curvature-frequency distribution of Nunnock River was not determined because LiDAR was unavailable for this site. The sub-catchment of Marshall Gulch had thicknesses to refusal rather than mobile regolith. This method may have resulted in overestimations of thicknesses and increased variability in thickness determination, but likely would not have significantly altered the TMR-curvature slope of 0.002 m$^2$ (Fig. 2c); curvature distributions were very broad indicating high curvature variability. As above, we used linear regression models of curvature and TMR, and we propagated vertical LiDAR uncertainty based on reported flight metadata for each cell according to the equations from Moore et al.[37] and Zevenbergen et al.[38] cited by Arc's curvature algorithm and divided by -100. Horizontal and vertical uncertainties were obtained through metadata provided on OpenTopography. Curvature uncertainty was calculated as standard error using the Method of Moments where we assumed correlation between uncertainty of neighbor and center cell points was 0 ($r = 0$). When vertical uncertainty was not provided, we assumed an uncertainty of 0.1 m (Supplementary Table 1). The effect of measurement scale on the TMR-curvature relationship in different environments merits further investigation because it defines the scale of TMR variation on the landscape. At present, combinations of spatially extensive TMR and high-resolution elevation data are rare (we have included all that we are aware of). These limitations will likely change as high resolution elevation data become increasingly available, though the labor of digging pits presents the main obstacle.

We performed independent tests of the model based on Eq. (1) on three catchments, Babbington Creek, Reynolds Mountain, and Gordon Gulch, using the standard deviations of curvature from all points of each catchment to estimate a TMR-curvature slope function. We estimated the intercepts from TMR observations on a planar surface at each catchment. Only a limited set of pits (6) were available for validation at Babbington compared to Reynolds Mountain where we had 8 pits. Two hundred and one (201) soil pits were available in Gordon Gulch, but pits were removed from this analysis if they fell within 10 m of rock outcrops and within areas that were recently glaciated, leaving 163 validation points.

**Data availability**. The datasets generated during and/or analyzed during the current study are available at BSU ScholarWorks [https://doi.org/10.18122/B2PM69][58]. Every sample at RC CZO is registered with an International Geo Sample Number through System for Earth Sample Registration (SESAR).

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

## Acknowledgements

This study was conducted in collaboration and cooperation with the USDA Agriculture Research Service, Northwest Watershed Research Center, Boise, Idaho, and landowners within the Reynolds Creek Critical Zone Observatory (RC CZO). Support for this research was provided by the NSF via RC CZO Cooperative agreement NSF EAR-1331872 (Lohse) and NSF EAR-1349384 (Crosby). Gordon Gulch data collection was funded by the Keck Geology Consortium, the National Science Foundation (NSF EAR-1062720), University of Connecticut Research Foundation, and NSF Boulder Creek Critical Zone Observatory (NSF EAR-072496). We thank Nicholas Howden and Dewayne Derryberry for providing statistical insights.

## Author contributions

K.A.L. and M.S.S. conceived the idea of examining topographic controls on soil thickness and carbon, and N.R.P. conceived of the idea of examining the relationships between catchment curvature and TMR-curvature function and cross-site comparison. N.R.P. and

K.A.L. collected field data and analyzed it and compiled cross-site comparison data along with S.E.G. and B.T.C.; K.A.L. and N.R.P. wrote the first paper draft. K.A.L., S.E.G., M.S. S., B.T.C., and N.R.P. edited and commented on the manuscript and contributed to later iterations.

## Additional information

**Competing interests:** The authors declare no competing interests.

