## [Peer Review File · Nature Communications]

Reviewers' comments:

Reviewer #1 (Remarks to the Author):

Review of Patton et al

This manuscript combines new data on soil thickness with previously measured data to argue that soil thickness can be predicted using topographic curvature. There have been papers in the past comparing curvature to erosion rates and soil thicknesses, but I do think this contribution is novel because:

- i) it presents new data that has been carefully collected
- ii) it tests a regression across a wide range of catchments
- iii) it highlights how simple spatial averaging or kriging of soil thickness maps are totally inadequate for predicting soil thickness.

I think the paper will be of interest to general audiences. I do have some concerns that I have detailed in lined comments below. One slightly pedantic point, but important, is that the authors have inverted the convention of calling ridgetops convex (G.K. Gilbert's seminal 1909 paper was called "The Convexity of Hilltops"): I really think the authors should change this. The authors also use the "negative curvature convention" which I find a totally unhelpful "convention" inexplicably introduced by ESRI that violates the mathematical definition of curvature. I could not live with myself if I did not raise a fuss about this and ask the authors to change it.

In addition, I think it is somewhat puzzling that the authors have not referred to past theory that could link curvature to soil thickness: theoretical predictions suggest that in the convex parts of the landscape the soil thickness should be proportional to the natural logarithm of curvature. It is fine if the data do not support this (in fact that would be an additional selling point of the novelty of the paper: the data suggest something is wrong with the theory!) but I suggest the theory should at least be mentioned, or better yet this relationship tested on the convex portions of the landscape.

Finally, I have a major concern that the curvature values are very, very high compared to past reported curvature values and I think there might be a problem with the calculations. I suspect this is a very simple matter of not dividing by grid spacing, but the authors must confirm the numbers are correct before this paper moves on to the next stage of review.

My overall impression is that there are some interesting findings here but I recommend some additional treatment of the data before the paper is really convincing. My lined comments are below.

Lined comments:

Line 13: In line 12 "soil thickness" was used. It should be used again here instead of "soil depth" for consistency. Also line 19. This is a bit of a hobby horse for me but I think soil thickness is correct since soil depth to me indicates a spatial variable in the vertical direction (e.g., carbon content is a function of soil depth).

Line 40: "z excluded hereafter" The Δ^2 operator is a mathematical operator so I think it is confusing, at best, to use this symbol combination as curvature. Why not just use "C"?

Line 66: "Negative curvature convention": This "convention" is simply incorrect. If elevation increases upwards (and it does) then convex areas have negative curvature. Why would you arbitrarily change the sign of a mathematical function? Just because ArcMap inexplicably uses the wrong sign for curvature does not mean anyone else should do so.

Line 80: I would note that this is the case by definition in linear regression, so that readers do not get the impression that the 0 m^{-1} was chosen arbitrarily.

Line 87: I don't really see how one would tell if there is an increase in variability or not between convex and concave locations from figure 1b. Perhaps the authors could quantify this?

Line 88: "Convex hollows". This is confusing. Valleys and hollows are in the concave portions of the landscape. Ridges and noses are convex. Gilbert's 1909 paper was on the "convexity" of hillslopes. If we were being pedantic we would say "convex up" but I would say that ridgetops being convex is conventional nomenclature in hillslope studies.

Line 100: "A standard deviation of 2.09 m^{-1} ". This is an extremely high variation in curvature.

A browse through a number of papers that report curvature typically have curvature numbers on the order of 0.01 m^{-1} . I am rather concerned that something has gone wrong in the calculations. Perhaps the curvatures are not divided by the grid spacing?

Line 108: Symbol $SD\{\text{del}^2\}$: Again, I'm not a fan of this symbology. The del squared operator is just that: a mathematical operator. So it isn't a good choice for the symbol of a variable. Why not use the standard Greek letter sigma for the standard deviation with the subscript C for curvature?

Line 110: I offer an alternative hypothesis: Coos bay is made up mostly of critical, planar hillslopes. Meaning that even if there are variations in erosion rate (which links to soil thickness, at least according to soil production theory), you would still get very similar values of curvature for the same soil thickness. The broad range of curvatures here is a function of the rapid erosion, which results in sharp ridges and incised valleys.

Line 131: Typo: h should be italic.

Line 132: I am happy that the data seems to follow a linear trend in some of the sites. But it is particularly interesting that it seems not to follow the predictions from theory. Theory states that soil thickness should go as the natural logarithm of curvature on convex hillslopes that are gentle. This comes from the steady state predictions that

$$(1) E \sim -D C$$

Where D is diffusivity and C is curvature (e.g., Roering et al., 2007 EPSL), and

$$(2) E \sim W \exp(-h/\gamma)$$

where W and gamma are empirical parameters (from Arjun's papers). Equation (1) above is only valid on gentle hillslopes because on steep hillslopes the relationship between erosion rate and curvature breaks down as the slopes become planar and tend to 0 curvature (see Roering et al 1999 and 2008). That is why in the Hurst et al papers we focused on hilltops. I wonder if you would get a logarithmic trend to the data that has negative curvature (no negative curvature convention! I mean the convex bits) after you eliminated parts of the landscape with steep slopes (i.e., > 0.4).

Line 171-172: This is a really important point. Soil maps based on kriging pedon data are going to be absolutely terrible at predicting soil thickness. I don't think I've read this anywhere before. This point is sufficiently important to merit an appearance in the abstract.

Line 190 and a number of places earlier: "roughness" is one of these woolly terms that means completely different things to different people. It seems the authors mean the standard deviation of topographic curvature in this instance. That either needs to be made very clear, or perhaps replace "roughness" with what you are actually measuring (i.e., curvature variability).

Line 338: Typo. Should be m^{-1} .

Line 343: I suggest the authors clarify this. I know of several other soil thickness datasets, including two we collected (DOI: 10.1002/esp.3754 and doi:10.1038/srep34438) and I believe Jon Pelletier has reported a few others. However these do tend to be limited to the convex portion of the landscape (like in the Gabet et al paper doi:10.1002/esp.3754) or the concave portion of the landscape (like the Parker et al paper doi:10.1038/srep34438). So I am okay with the limited data sets the authors have chosen but it would be useful just to be more specific about what they mean by "spatially extensive".

Figure 2: I wonder if this is an artefact of grid spacing. In rapidly eroding landscapes you should get sharp ridges. You lose more curvature information in coarse grids if the ridges are sharp (see section 5.1.1 of Grieve et al 2016 doi:10.5194/esurf-4-627-2016). If ridges are sharp there will be a greater range of curvatures. So I think a grid resolution bias would result in a lower slope of TMZ-curvature function at higher standard deviation of curvature values. Are the authors able to test the relationship on different grid resolutions? If not I would offer readers the alternative hypothesis that in landscapes with very sharp ridges you might lose some curvature information. In fact, the data in Table 2 seems to show this effect.

S. Mudd

Reviewer #2 (Remarks to the Author):

Review of Predicting soil thickness on soil mantled hillslopes

The authors have correctly identified that one of the most important missing data inputs for environmental modelling is soil thickness. They correctly identify the broad range of applications where the science could be significantly improved with better maps of soil depth. That said they are not the first to have had that insight.

They have done a reasonable job of identifying the work by geomorphologists in this area. They have missed the work of Freer et al., 2002 and Saco et al. 2006 who made the case that the bedrock topography was disconnected from the surface topography, so that at least at the fine scale the soil thickness (i.e. surface elevation – bedrock elevation) has a strong component based on the bedrock topography (but not the surface topography), which in turn can be explained by the internal dynamics of the physics of the combination of the spatial distribution of the soil moisture and the soil production function.

However, while doing a generally good job of recognising the contribution of geomorphology to the topic of predicting soil depth they have done a poor job of recognising the work by the soil science community, particularly that from the digital soil mapping DSM subcommunity, and those working on the international Global Soil Map initiative. This is not only a question of recognition of prior art but the comparison by the authors of their method with standard kriging (lines 171-181) is misleading. Nobody in DSM would suggest that standard kriging would do particularly well and the standard technique used is regression Kriging where the environmental variables such as wetness index, elevation, longitudinal concavity (and surface roughness as used by the authors of this paper) are used as the independent variables in the geostatistical interpolation (see, for example, Hengl et al., 2004; Kuriakose et al., 2009; Taylor et al., 2013; Shangguan et al., 2016. There are others but these are the ones that are easy to find in my Endnote database).

These deficiencies notwithstanding the main problem with the paper is the site specific nature of the analysis, and the lack on any insight into how the results from their study site might be transferred to another site. They present a model that uses surface curvature as the main explanatory variable for soil depth, and then assert that this is the best relationship. However, other than the flawed Kriging analysis they have not demonstrated that other feasible relationships (e.g. using other explanatory variables found in the literature, for example as listed in the papers above from the DSM community) yield inferior predictors. Thus no case is made that the relationship presented in Equations (1) is anything other than one of many potential feasible predictors of soil depth. More specifically, Kurikose et al. 2009 has surface curvature as one of their ten tested variables, but it is not significant in their regression Kriging (their Table 3). Surface curvature is only the third most significant variable in the analysis of Taylor et al. 2013 (their Figure 4). Shangguan et al. 2016 doesn't find curvature significant at all, through curvature is correlated with MrVBF which was mid-range in their table of explanatory values (their Figure 6).

All of these studies (including the authors') are regressions between soil depth data and independent variables so do not provide causal relationships. Without some form of causal relationship it is rather speculative to suggest that Equation (1) is universally applicable (in fact their Figure 2A where they examine other independent sites shows considerable scatter using their relationship suggesting that it is not transferable). Other studies have consistently shown that the strongest independent predictor of soil depth is vegetation. The rationale for this is that vegetation growth is, at least for drier regions, limited by water availability and the deeper the soil the greater is the water storage capacity of the soil. This is not a causal relationship but it is a predictor. An early work that links soil depth and vegetation vigour is Knorr and Lakshmi (2001), and there is now a significant body of research on this to underpin better modelling of latent heat energy exchanges between the land surface and the atmosphere in climate models. Moreover I

recall a paper a few years back, but can't seem to find it in my database, that went beyond a simple regression relationship and that coupled remote sensing of vegetation vigour (LAI/NDVI as I recall) with a soil moisture model to estimate soil profile water holding capacity and indirectly soil depth. In principle THIS IS transferable to other catchments, yet vegetation does not rate a mention in the paper being reviewed.

In conclusion while this is an interesting paper to add to the database of available soil depth data I don't see that it adds significantly to the state-of-the-art, and due to its style of presentation and ignorance of prior art in the DSM community may actually impair progress. I recommend that this paper be rejected.

Freer, J., J. J. McDonnell, K. J. Beven, N. E. Peters, D. A. Burns, R. P. Hooper, B. Aulenbach, and C. Kendall (2002), The role of bedrock topography on subsurface storm flow, *Water Resources Research*, 38(12), art. no.-1269.

Hengl, T., G. B. M. Heuvelink, and A. Stein (2004), A generic framework for spatial prediction of soil variables based on regression-kriging, *Geoderma*, 120(1-2), 75-93.

Knorr, W., and V. Lakshmi (2001), Assimilation of fAPAR and surface temperature into a land surface and vegetation model, in *Land surface hydrology, meteorology and climate: Observations and modeling*, edited by V. Lakshmi, J. Albertson and J. Schaake, pp. 177-200, American Geophysical Union, Washington DC.

Kuriakose, S. L., S. Devoka, D. G. Rossiter, and V. G. Jetten (2009), Prediction of soil depth using environmental variables in an anthropogenic landscape, a case study in the Western Ghats of Kerala, India, *Catena*, 79, 27-38, doi:10.1016/j.catena.2009.05.005.

Saco, P. M., G. R. Willgoose, and G. R. Hancock (2006), Spatial organization of soil depths using a landform evolution model, *Journal of Geophysical Research (Earth Surface)*, 111, F02016, doi:10.1029/2005JF000351.

Shangguan, W., T. Hengl, J. Mendes de Jesus, and Y. Dai (2016), Mapping the global depth to bedrock for land surface modeling, *Journal of Advances in Modeling Earth Systems*, 9, 65-88, doi:10.1002/2016MS000686.

Taylor, J. A., F. Jacob, M. Galleguillos, T. Prévot, N. Guix, and P. Lagacherie (2013), The utility of remotely-sensed vegetative and terrain covariates at different spatial resolutions in modelling soil and watertable depth (for digital soil mapping), *Geoderma*, 193-194, 83-93, doi:10.1016/j.geoderma.2012.09.009.

Responses to reviewers:

We are grateful for the opportunity to submit a revised version of our manuscript and hope the Editor and Reviewers agree that the changes made in response to Reviewer comments have greatly increased the clarity, context, and utility of the paper. We apologize on our delayed response, but based on the 2 reviews, especially those of reviewer 1, we re-analyzed the data to remove the negative curvature convention and adjust ArcGIS outputs by -100 to allow for direct comparison to the current literature. This required substantial amount of time to redo analyses, remake figures and revise the associated text. The lead author also started a PhD program in Australia, and this transition resulted in further delays. In addition to these revisions, we followed reviewer 1's suggestion and tested the prevailing assumption that relates soil thickness to the natural logarithm of curvature. Our new analyses (new Supplementary Fig. 4) lend more support to our finding of TMR varying linearly rather than in proportion to the natural logarithm of curvature. These results are in conflict with the current theoretical paradigm raising questions with regards to underlying assumptions of steady state soil thickness, the linear relation between slope and sediment flux and/or the exponential formulation of the soil production model. The linear TMR-curvature relationship is also advantageous over the natural logarithm because it can handle both positive and negative values for curvature. Collectively, our findings indicate that our simple linear TMR-curvature model may produce more robust estimates of TMR across the full range of curvature values than assumptions using a natural logarithm relation as well as produce just as reliable estimates of TMR as kriging-based interpolations with significantly less labor and cost. We respond to reviewers' general and specific comments in bold.

Reviewers' comments:

Reviewer #1 (Remarks to the Author):

This manuscript combines new data on soil thickness with previously measured data to argue that soil thickness can be predicted using topographic curvature. There have been papers in the past comparing curvature to erosion rates and soil thicknesses, but I do think this contribution is novel because:

- i) it presents new data that has been carefully collected
- ii) it tests a regression across a wide range of catchments
- iii) it highlights how simple spatial averaging or kriging of soil thickness maps are totally inadequate for predicting soil thickness.

I think the paper will be of interest to general audiences. I do have some concerns that I have detailed in lined comments below. One slightly pedantic point, but important, is that the authors have inverted the convention of calling ridgetops convex (G.K. Gilbert's seminal 1909 paper was called "The Convexity of Hilltops"): I really think the authors should change this.

Response: Thank you for pointing this out. We recognize that we choose an uncommon convention and have inverted it in our revisions to ensure that readers, especially those familiar with Gilbert, will not be confused.

The authors also use the “negative curvature convention” which I find a totally unhelpful “convention” inexplicably introduced by ESRI that violates the mathematical definition of curvature. I could not live with myself if I did not raise a fuss about this and ask the authors to change it.

Response: We appreciate this suggestion, and like the reviewer, we have considered this. Although the “negative curvature convention” is widely used and accepted within the ArcGIS community, it is also widely rejected and challenged. But, like the reviewer has stated, this convention violates the mathematical definition of curvature, so we have decided remove the “negative curvature convention” as suggested. We have revised our manuscript's text and data to address your comments

“...hillslope curvature^{2,5, 21} (C), quantified as the rate of change in slope from a fixed point on a landscape in all directions.” (line 42)

“ Curvature was calculated as the rate of change in slope from a fixed point relative to eight neighboring cells^{36,37} using a geographical information system (ArcGIS v.10.2.2, ESRI, Redlands, CA). We utilized ArcGIS primary curvature output, which is derived from Zevenberger and Thorne³⁸ and Moore et al.³⁷ equations; however, we removed the negative curvature convention and divided the output by 100 to allow for direct comparison to current literature.” (line 72)

“We used a linear regression model with curvature and TMR. We propagated vertical LiDAR uncertainty based on reported flight metadata for each cell according to the equations from Moore et al.³⁷ and Zevenbergen et al.³⁸ cited by Arc’s curvature algorithm. To be consistent with current literature, we removed the negative curvature and percent convention, as used by ArcGIS, by dividing curvature values by -100.” (line 348)

In addition, I think it is somewhat puzzling that the authors have not referred to past theory that could link curvature to soil thickness: theoretical predictions suggest that in the convex parts of the landscape the soil thickness should be proportional to the natural logarithm of curvature. It is fine if the data do not support this (in fact that would be an additional selling point of the novelty of the paper: the data suggest something is wrong with the theory!) but I suggest the theory should at least be mentioned, or better yet this relationship tested on the convex portions of the landscape.

Response: Thank you for your insightful comment. We agree that the link between past theoretical soil thickness predictions and our work would be a valuable contribution to our manuscript. In order to compare our model with the prevailing one, we extracted the convex regions of each field site and evaluated whether the natural logarithm of curvature was a better fit than a linear relationship. As seen in new Supplementary Figure 4, we find that the linear TMR-curvature relationship fits better than the logarithmic relationship. Additional text has been added to incorporate this significant contextual connection to previous work:

“Assuming sediment flux is linearly proportional to slope, conservation laws predict an inverse linear relation between the soil production rate and hillslope curvature^{2,5, 21} (C), quantified as the rate of change in slope from a fixed point on a landscape in all directions.

Researchers have also independently demonstrated that soil production rates decrease exponentially with increasing soil thickness^{2,3,20-22,29,30}. Combining these two relationships establishes soil thickness as proportional to the natural logarithm of curvature^{21, 26}, explicitly on convex, divergent hillslopes (associated with negative curvatures). In contrast to convex areas, concave, convergent areas have received less attention in part because soil production is reduced under thick soil cover and theoretical predictions using the natural logarithm of curvature fail for positive values.” (line 40)

“We also evaluated model selection within the convex, negative curvature regions and found more support for TMR varying linearly ($N=22$, $r^2=0.63$, $RMSE=0.18$) rather than in proportion to the natural logarithm of curvature ($r^2=0.37$, $RMSE=0.22$). This is in conflict with the current theoretical paradigm and may result from failed assumptions surrounding steady state soil thickness, the linear relation between slope and sediment flux or the exponential formulation of the soil production model. The linear TMR-curvature relation is also advantageous over the natural logarithm because it can handle both positive and negative values for curvature.” (line 94)

“We anticipated that TMR-curvature relations would exhibit greater variation at larger curvatures (concave, convergent areas) owing to either over-thickening or recently failed soils in zero-order hollows. However, there was no change in variability with curvature at our site, evidenced by homogeneity in variance around the best-fit line (Fig 1a). These findings suggest that the soil thickness in concave hollows of Johnston Draw are regulated by frequent and efficient transport processes instead of erratic evacuations as suggested by Dietrich et al.⁵. Instead of thickening indefinitely, hollows may be maintained by creep or surface erosion. Similarly, TMR thicknesses on concave toe slopes on terraces or floodplains may be regulated by creep or surface erosion rather than evacuation by lateral channel migration. This supposition is consistent with findings by Dietrich et al.⁵ who found predicted soil thicknesses greater than 1 m on toe slopes, but warrants further study in this and other landscapes to understand the processes underlying this linear relationship.” (line 102)

“Indeed, our analyses showed more support for TMR varying linearly rather than in proportion to the natural logarithm of curvature (Supplementary Fig. 4) raising questions with regards to underlying assumptions of steady state soil thickness, the linear relation between slope and sediment flux and/or the exponential formulation of the soil production model.” (line 209)

Finally, I have a major concern that the curvature values are very, very high compared to past reported curvature values and I think there might be a problem with the calculations. I suspect this is a very simple matter of not dividing by grid spacing, but the authors must confirm the numbers are correct before this paper moves onto the next stage of review. My overall impression is that there are some interesting findings here but I recommend some additional treatment of the data before the paper is really convincing.

Response: We understand the reviewer’s concerns that our curvature datasets appear very high. We want to assure them that we are using the curvature algorithm derived from Zevenbergen and Thorne (1987) and Moore et al. (1991). The reason our reported values are high is that we used ArcGIS’ primary output, which quantifies slope as a percentage rather than as a gradient (rise over run), resulting in values 100 times greater than expected (see URL below). However, we have removed both ArcGIS’s negative

curvature convention and scaled the values down by 100 as requested in above comments. Thus, the values are now congruent with those in other publications.

<http://desktop.arcgis.com/en/arcmap/10.3/tools/spatial-analyst-toolbox/how-curvature-works.htm>

Lined comments:

Line 13: In line 12 “soil thickness” was used. It should be used again here instead of “soil depth” for consistency. Also line 19. This is a bit of a hobby horse for me but I think soil thickness is correct since soil depth to me indicates a spatial variable in the vertical direction (e.g., carbon content is a function of soil depth).

Response: Thank you for the suggestion. We have replaced all instances of “soil depth” with “soil thickness” for consistency.

Line 40: “z excluded hereafter” The Δ^2 operator is a mathematical operator so I think it is confusing, at best, to use this symbol combination as curvature. Why not just use “C”?

Response: Thank you for the suggestion. For simplicity, we have adopted the “C” notation as you suggested, which is broadly consistent with Hurst et al. 2012 and others.

Line 66: “Negative curvature convention”: This “convention” is simply incorrect. If elevation increases upwards (and it does) then convex areas have negative curvature. Why would you arbitrarily change the sign of a mathematical function? Just because ArcMap inexplicably uses the wrong sign for curvature does not mean anyone else should do so.

Response: We removed this convention. See response above.

Line 80: I would note that this is the case by definition in linear regression, so that readers do not get the impression that the 0 m^{-1} was chosen arbitrarily.

Response: We have updated the sentence to read, “*Note that for this relationship, the y-intercept is defined as the value when $x=0$; in this case, the y-intercept is the TMR on a planar surface where the curvature is 0 m^{-1} .*” (line 89)

Line 87: I don’t really see how one would tell if there is an increase in variability or not between convex and concave locations from figure 1b. Perhaps the authors could quantify this?

Response: We think that the reviewer is referring to Figure 1a as referenced in the manuscript at this line. An increase in variability with curvature would manifest as a trend (with curvature) in the size of the residuals from the best-fit line. We do not explicitly plot the residuals vs. curvature in the figure due to space constraints, but include that plot here for confirmation of this observation. We have modified this text to support this idea by stating:

“We anticipated that TMR-curvature relations would exhibit greater variation at larger curvatures (concave, convergent areas) owing to either over-thickening or recently failed soils in zero-order hollows. However, there was no change in variability with curvature at our site,

evidenced by homogeneity in variance around the best-fit line (Fig 1a).” (line 102)

Line 88: “Convex hollows”. This is confusing. Valleys and hollows are in the concave portions of the landscape. Ridges and noses are convex. Gilbert’s 1909 paper was on the “convexity” of hillslopes. If we were being pedantic we would say “convex up” but I would say that ridgetops being convex is conventional nomenclature in hillslope studies.

Response: As noted above, we have modified the text to avoid this confusion and it now reads: “These findings suggest that the soil thickness in concave hollows of Johnston Draw are regulated by frequent and efficient transport processes instead of erratic evacuations as suggested by Dietrich et al.⁵. ” (line 107)

Line 100: “A standard deviation of 2.09 m^{-1} ”. This is an extremely high variation in curvature. A browse through a number of papers that report curvature typically have curvature numbers on the order of 0.01 m^{-1} . I am rather concerned that something has gone wrong in the calculations. Perhaps the curvatures are not divided by the grid spacing?

Response: We have addressed this comment in above text. In short, we were using a percent slope instead of a ratio. This is now fixed.

Line 108: Symbol $SD\{\Delta^2\}$: Again, I’m not a fan of this symbology. The del squared operator is just that: a mathematical operator. So it isn’t a good choice for the symbol of a variable. Why not use the standard Greek letter sigma for the standard deviation with the subscript C for curvature?

Response: We have adopted this suggested convention and have changed “SD (Δ^2)” to “Sigma with subscript C” throughout.

Line 110: I offer an alternative hypothesis: Coos bay is made up mostly of critical, planar hillslopes. Meaning that even if there are variations in erosion rate (which links to soil thickness, at least according to soil production theory), you would still get very similar values of curvature for the same soil thickness. The broad range of curvatures here is a function of the rapid erosion, which results in sharp ridges and incised valleys.

Response: Your hypothesis works in some ways. We agree that steep, mass movement-dominated hillslopes are expected to be composed of near-planar hillslopes (zero

curvature) at some critical slope determined by substrate strength. On these threshold, planar slopes (as well as in debris flow dominated hollows), soil thickness will vary depending on ‘elapsed time since last failure,’ and recent disturbances such as tree throw or burrowing rather than just curvature. We want to emphasize that the Coos Bay thickness measurements in our Figure 2a come only from convex noses and ridges, away from sharp inflections or debris flow impacted hollows. In contrast, Figures 2b and 2c that show a broad distribution of curvatures are extracted from the watershed DEM, a much larger domain than where thickness was measured.

We want to emphasize that even though Heimsath et al. (2001, ESPL) recognizes that mass movements occur at the Coos Bay site, all soil thickness measurements were made on convex noses and ridges, away from sharp inflections or debris flow impacted hollows (see their Figure 4a,b). Any concave values in Figure 2a came from local depressions on otherwise convex landforms. They state, “we specifically avoided the planar and steep slopes to remain out of the region where the non-linear model [inclusive of mass movements] may predict sediment flux more closely than the linear model [relating flux to slope].”

They conclude that the poor correlation between thickness and curvature is a consequence of stochastic soil formation and transport processes. They state, “Field observations on the nature of soil production and removal show episodic processes of tree-throw, animal burrowing and shallow landsliding operating across the landscape. These processes lead to highly variable local soil depths over time and measurements of soil depth may only reflect an instantaneous snapshot of the soil depths across the landscape, rather than the long-term, steady-state soil thickness assumed for our nuclide interpretations.” In essence, stochastic disturbances (even on convex features free of mass movements) result in BOTH an irregular surface and an irregular soil-rock interface, complicating correlations between local curvature and soil thickness.

To clarify this, we have modified the text to read:

“Sites with high σ_c (e.g. Marshall Gulch and Coos Bay) had TMR-curvature slopes near zero indicating that curvature poorly predicted TMR. In these sites, large magnitude, high frequency stochastic disturbances (e.g. tree-throw, mass movements and burrowing) alter both the surface topography and the thickness of mobile regolith^{21,27}. In contrast, sites with low σ_c (e.g. Johnston Draw and Point Reyes) had both high TMR-curvature slopes and r^2 values. In these catchments, soil formation and transport processes are likely driven by low-magnitude, gradual processes (rheologic creep, lesser bioturbation) that smooth out the surface topography, resulting in curvature values that explain much of the variation in TMR (Fig. 2a).” (line 132)

Line 131: Typo: h should be italic.

Response: Thank you, we have addressed this typo.

Line 132: I am happy that the data seems to follow a linear trend in some of the sites. But it is particularly interesting that it seems not to follow the predictions from theory. Theory states that soil thickness should go as the natural logarithm of curvature on convex hillslopes that are gentle. This comes from the steady state predictions that

$$(1) E \sim -D C$$

Where D is diffusivity and C is curvature (e.g., Roering et al., 2007 EPSL), and

$$(2) E \sim W \exp(-h/\gamma)$$

where W and γ are empirical parameters (from Arjun's papers). Equation (1) above is only valid on gentle hillslopes because on steep hillslopes the relationship between erosion rate and curvature breaks down as the slopes become planar and tend to 0 curvature (see Roering et al 1999 and 2008). That is why in the Hurst et al papers we focused on hilltops. I wonder if you would get a logarithmic trend to the data that has negative curvature (no negative curvature convention! I mean the convex bits) after you eliminated parts of the landscape with steep slopes (i.e., > 0.4).

Response: As discussed above, the vast majority of our data come from convex, divergent hilltops and noses. Our own data from Johnson Draw is more broadly distributed across concave and convex regions. Even when we eliminate the steeper and concave measurements, the linear fit is still better than a logarithmic one, see Supplementary Fig. 4.

Line 171-172: This is a really important point. Soil maps based on kriging pedon data are going to be absolutely terrible at predicting soil thickness. I don't think I've read this anywhere before. This point is sufficiently important to merit an appearance in the abstract.

Response: Thank you for your comments; however, at the request of the 2nd reviewer, we ran a regression kriging model for soil thickness and also re-ran our model. We found that the RMSE for the TMR-curvature approach had a typo in the previous submission, and the regression kriging and TMR-curvature approaches now produce similar results. Specifically, we previously reported a RMSE of 0.2 m for the TMR-curvature approach, but the correct value is 0.44 m (see table below). Although this initially might seem disappointing, the method still represents a massive savings in labor and cost because it only requires a single pit on a planar hillslope. The revised text in the manuscript now reads:

“Perhaps more importantly, the TMR-curvature model, based on a 5 m resolution LiDAR to estimate σ_c and one soil profile on a planar surface to estimate the intercept, performs just as well as kriging-based interpolations. Indeed, comparison of our approach to simple, ordinary, and regression kriging models at Gordon Gulch shows that kriging models do not improve TMR estimates compared to our TMR-curvature model ($r^2=0.19$ and RMSE of ~ 0.44 m compared to $r^2 < 0.02$ and RMSE of ~ 0.4 m for both simple and ordinary kriging and $r^2=0.06$ and RMSE of ~ 0.37 m for regression kriging). We only conducted this comparison at Gordon Gulch where there were sufficient samples to compare techniques⁴⁵ (113 build and 50 test pits, 163 total).” (line 199)

Discussion

“Findings from our study indicate that our linear TMR-curvature model may produce just as reliable estimates of TMR as kriging-based interpolations with significantly less labor and cost. Our model also provides more robust estimates of TMR across the full range of curvature values than using a natural logarithm relation. Indeed, our analyses showed more support for TMR varying linearly rather than in proportion to the natural logarithm of curvature (Supplementary Fig. 4) raising questions with regards to underlying assumptions of steady state soil thickness, the linear relation between slope and sediment flux and/or the exponential formulation of the soil production model.” (line 209)

In addition, we have included language in our abstract to point out that our simple model provides similar results compared a more complex model that requires significantly larger sample size:

“This provides a simple empirical model for predicting the spatial distribution of soil thickness in a variety of catchments based only on high-resolution elevation data and few soil profiles. Despite our model’s simplicity, the results are comparable to more complex kriging methods that require significantly more parameterization and field measurements.” (line 17)

	Predicted versus Measured from Validation Set (n=50)		
Method	R ²	Slope	RMSE (m)
TMR-Curvature (n=1)	0.19	0.37	0.44
Simple and Ordinary Kriging (n=113)	< 0.02	< 0.07	~0.4
Regression Kriging (n=113)	0.06	0.71	0.37

Line 190 and a number of places earlier: “roughness” is one of these woolly terms that means completely different things to different people. It seems the authors mean the standard deviation of topographic curvature in this instance. That either needs to be made very clear, or perhaps replace “roughness” with what you are actually measuring (i.e., curvature variability).

Response: We agree, and although different groups use the term differently, we use “roughness” consistently. However, to clarify the ambiguity of this term we have added addition text for clarification:

“In contrast to the observation that curvature distributions were normal and centered on 0 m⁻¹ for all catchments, surface roughness, defined here as the standard deviation in catchment curvature (σ_c) at a given scale³⁹, varied from 0.0209 to 0.0713 m⁻¹ across sites (Table 1, Fig. 2c).” (line 127).

“Consistent with this idea, one study showed that catchment roughness as measured by the standardized topographic position index can be used as a proxy for sediment availability⁴² suggesting that catchment surface roughness may have some utility as a proxy for mapping out different geomorphic processes or process rates.” (line 145)

Line 338: Typo. Should be m⁻¹.

Response: Thank you, we have addressed this typo.

Line 343: I suggest the authors clarify this. I know of several other soil thickness datasets, including two we collected (DOI: 10.1002/esp.3754 and doi:10.1038/srep34438) and I believe Jon Pelletier has reported a few others. However these do tend to be limited to the convex portion of the landscape (like in the Gabet et al paper doi:10.1002/esp.3754) or the concave portion of the landscape (like the Parker et al paper doi:10.1038/srep34438). So I am okay with the limited data sets the authors have chosen but it would be useful just to be more specific about what they mean by “spatially extensive”.

Response: The datasets provided are appreciated however they are unfortunately not appropriate because Gabet et al., 2015 and Parker et al., 2016 collected soil thickness

using a tile probe and were limited to specific regions of the watershed as the reviewer explained. We have included additional text to explain the importance of excavating soil pits:

“Studies that dig to refusal, bedrock or at a predetermined depth may include or exclude portions of the mobile regolith resulting in inconsistencies. In addition, methods such as augering, soil tile pole, and knocking pole method may underestimate thickness due to highly rocky layer and/or overestimate depth due to fractures in bedrock. Without properly identifying the mobile regolith boundary with soil pits, the likelihood of producing topographic relationships is low. For this reason, comparisons of our work to many other studies that have used these other methods^{46 47} are difficult and require considerable knowledge of both the research methods and site location.” (line 254)

Figure 2: I wonder if this is an artefact of grid spacing. In rapidly eroding landscapes you should get sharp ridges. You lose more curvature information in coarse grids if the ridges are sharp (see section 5.1.1 of Grieve et al 2016 doi:10.5194/esurf-4-627-2016). If ridges are sharp there will be a greater range of curvatures. So I think a grid resolution bias would result in a lower slope of TMZ-curvature function at higher standard deviation of curvature values. Are the authors able to test the relationship on different grid resolutions? If not I would offer readers the alternative hypothesis that in landscapes with very sharp ridges you might lose some curvature information. In fact, the data in Table 2 seems to show this effect.

Response: Thank you for your comment. The reviewer is correct that the grid size plays an important role in our analyses. The purpose of Table 2 was indeed to show the sensitivity of grid size to the TMR-curvature relationship. We state:

“We extracted curvature values for Johnson Draw from a LiDAR DEM resampled to 3 m resolution because a sensitivity analysis showed that this resolution provided the highest correlation between TMR and curvature (Table 2). When comparing Johnson Draw with other datasets, we resampled the LiDAR data to 5 m resolution because some of the cross-site datasets were manually collected at this resolution^{2,3,20-22}.” (line 76)

We also included this line in the main text of discussion

“Our sensitivity analysis at Johnston Draw indicates that the TMR-curvature relationship is highly sensitive to scale (Table 2), with some deterioration of the relationship with the 5 m resolution resampling and considerable deterioration at 30 m resolution, which is typical of widely available DEM’s. Our cross-site comparison using consistent TMR measurements and high resolution LiDAR data may explain why our findings have not been previously reported”. (line 264)

Finally, in the methods we state,

“A 3 m DEM was selected based on a sensitivity analysis of the TMR-curvature function to the resolution (scale) of the DEM (Table 2); the original 1 m DEM was resampled to 3, 5, 10, 20, 30 and 50 m DEM utilizing the mean elevation of the nine adjacent cells. The TMR-curvature correlation was strongest for curvature derived from the 3 m DEM ($N=38$, $r^2=0.86$, $RMSE=0.20$ m) whereas it deteriorated some with the 5 m resolution resampling ($N=38$, $r^2=0.44$, $RMSE=0.40$ m) and considerably at a resolution of 30 m ($N=38$, $r^2=0.25$, $RMSE=0.47$ m). For local estimates of soil thickness (Fig. 1b), the 3 m DEM was used.” (line 318)

Reviewer #2 (Remarks to the Author):

Review of Predicting soil thickness on soil mantled hillslopes

They have missed the work of Freer et al., 2002 and Saco et al. 2006 who made the case that the bedrock topography was disconnected from the surface topography, so that at least at the fine scale the soil thickness (i.e. surface elevation – bedrock elevation) has a strong component based on the bedrock topography (but not the surface topography), which in turn can be explained by the internal dynamics of the physics of the combination of the spatial distribution of the soil moisture and the soil production function.

Response: We appreciate your comments and agree that the work by Freer et al., 2002 and Saco et al., 2006 are important and insightful; however, their findings are difficult to incorporate into our study for several key reasons. First, the methods used to determine “soil thickness” differed: Freer et al. (2002) using a 2.54-cm soil corer or auger (for deeper soils) to refusal. This method can lead to biased soil thickness because of coarse material in the subsurface that may appear to be immobile regolith, but in fact, is merely a cobble or boulder. We added additional text to emphasize this point:

“Studies that dig to refusal, bedrock or at a predetermined depth may include or exclude portions of the mobile regolith resulting in inconsistencies. In addition, methods such as augering, soil tile pole, and knocking pole may underestimate thickness in highly rocky layers and/or overestimate depth due to fractures in bedrock. Without properly identifying the mobile regolith boundary with soil pits, the likelihood of producing topographic relationships is low. For this reason, comparisons of our work to other studies that used these other methods^{46 47} are problematic and require considerable knowledge of both the research methods and site location.” (line 254)

Furthermore, both studies examine relationships between surface and subsurface topography at a 1-2 m grid size. In our sensitivity analysis (Table 2), we find that at the 1-2 m scale, the TMR-curvature relationship is relatively weak. This is most likely due to noise in the topographic data or perhaps may result from dominant geomorphic processes occurring at 3 m resolution or larger. As we state in our manuscript:

“We extracted curvature values for Johnson Draw from a LiDAR DEM resampled to 3 m resolution because a sensitivity analysis showed that this resolution provided the highest correlation between TMR and curvature (Table 2). When comparing Johnson Draw with other datasets, we resampled the LiDAR data to 5 m resolution because some of the cross-site datasets were manually collected at this resolution^{2,3,20-22}.” (line 76)

“A 3 m DEM was selected based on a sensitivity analysis of the TMR-curvature function to the resolution (scale) of the DEM (Table 2); the original 1 m DEM was resampled to 3, 5, 10, 20, 30 and 50 m DEM utilizing the mean elevation of the nine adjacent cells. The TMR-curvature correlation was strongest for curvature derived from the 3 m DEM ($N=38$, $r^2=0.86$, $RMSE=0.20$ m) whereas it deteriorated some with the 5 m resolution resampling ($N=38$, $r^2=0.44$, $RMSE=0.40$ m) and considerably at a resolution of 30 m ($N=38$, $r^2=0.25$, $RMSE=0.47$ m). For local estimates of soil thickness (Fig. 1b), the 3 m DEM was used.” (line 318)

Another important observation to note from Freer et al., 2002 and Saco et al. 2006 is the field area (Panola Mountain Research Watershed, Georgia) is surrounded by extensive rock outcropping that most likely affects the TMR-curvature relationship. We write:

“In these catchments, steep hillslopes and the abundance of trees can result in more frequent and spatially heterogeneous topographic disturbances such as landslides and tree throw that increase surface roughness and therefore influence the variability in TMR. In Johnston Draw, Marshall Gulch and Gordon Gulch, rock outcrops create similar local disturbances, features not captured in the mobile regolith model. As a consequence, pit locations within 10 m of rock outcrops were excluded from the analysis.” (line 230)

We hypothesize the differences in the TMR-curvature relationships could be related to different conditions for chemical weathering and thus explain some of the observed variability. In particular, Saco et al., 2006 provides an excellent mechanism for such regions within the landscape and to recognize this we have modified our text to include the strong interaction bedrock topography and chemical weathering.

“We posit that soil thicknesses in catchments with high surface roughness are not governed solely by local topography but rather by multiple geomorphic processes (e.g. mass movements, tree throw, etc.) and in-situ soil evolution that may influence soil production rates³³.” (lines 232)

However, while doing a generally good job of recognizing the contribution of geomorphology to the topic of predicting soil depth they have done a poor job of recognizing the work by the soil science community, particularly that from the digital soil mapping DSM subcommunity, and those working on the international Global Soil Map initiative. This is not only a question of recognition of prior art but the comparison by the authors of their method with standard kriging (lines 171-181) is misleading. Nobody in DSM would suggest that standard kriging would do particularly well and the standard technique used is regression Kriging where the environmental variables such as wetness index, elevation, longitudinal concavity (and surface roughness as used by the authors of this paper) are used as the independent variables in the geostatistical interpolation (see, for example, Hengl et al., 2004; Kuriakose et al., 2009; Taylor et al., 2013; Shanguan et al., 2016.)

Response: Thank you for your response and your concerns are appreciated and valid. We believe that there may be a misunderstanding in the goal of this work, which is to provide *“... a simple empirical model for predicting the spatial distribution of soil thickness in a variety of catchments based only on high-resolution elevation data and few soil profiles.”* (lines 17). It was not our intention to mislead the readers and undermine the extensive work by the Global Soil Mapping and Digital Soil Mapping communities, which we see as complementary to this effort, but operating at a different scale. Our work specifically utilizes high-resolution LiDAR-derived DEMs at resolutions between 3 and 5 m; this differs markedly from the resolutions of the DEMs in the cited work, which range between 10 and 6000 m. In fact, the one study at 10 m resolution by Taylor et al. (2013) finds that terrain attributes such as curvature are the best covariates for modeling soil depth. However, to insure we recognize the work within the Digital and Global Soil Mapping community we have modified the following text to read:

“As the number of sites with TMR measurements increase, our model will likely be complementary to the digital and global soil mapping community⁴⁸ such as allowing rigorous testing of global soil thickness models²⁶.” (line 276)

We hope that this manuscript may prompt further work to understand the variation in strength of the relationship with scale, and believe that scaling issues may have obscured the strength of the TMR-curvature relationship in the past. Although LiDAR is rapidly

becoming a staple in most geomorphic and environmental studies its accessibility was limited. We state in our in our manuscript:

“In contrast to TMR measurements, high-resolution elevation data are becoming increasingly abundant and technology associated with unmanned aerial vehicle, airborne and remotely sensed data collection is advancing rapidly and will likely provide sufficient elevation data on demand. Our sensitivity analysis at Johnston Draw indicates that the TMR-curvature relationship is highly sensitive to scale (Table 2), with some deterioration of the relationship with the 5 m resolution resampling and considerable deterioration at 30 m resolution, which is typical of widely available DEM’s. Our cross-site comparison using consistent TMR measurements and high resolution LiDAR data may explain why our findings have not been previously reported.” (line 261)

“These limitations will likely change as high resolution elevation data become increasingly available, though the labor of digging pits presents the main obstacle.” (line 399)

Our purpose for providing a comparison between simple and ordinary kriging methods with our method was based on the practical application of scientist within fields that do not use complex models. Utilizing other variables in addition to curvature is a good suggestion but is beyond the scope of this research. We expanded to include a regression kriging model based on the model data set from the Gordon Gulch (113 model sites and 50 validation sites). The regression kriging model used hillslope curvature as an environmental parameter to predict soil thickness. We utilized the same relationship in our manuscript (TMR-curvature) to inform the regression kriging semivariogram, and subsequently universal kriging was used to interpolate soil thickness predictions.

The regression kriging model performs slightly better than the TMR-curvature model (see table below), but requires substantially more time and money. Our method only used one pit at a planar surface and the standard deviation of catchment curvature derived from LiDAR rather than many pits that are typically required to successfully krige with a similar uncertainty. With this in mind, we have updated the text as described in response to Reviewer #1 (above) to highlight the success of both models, and what we still believe are important contributions of the TMR-curvature approach. The real benefit of our work is its simplicity and accessibility for researchers to use in a variety of fields and disciplines.

	Predicted versus Measured from Validation Set (n=50)		
Method	r ²	Slope	RMSE (m)
TMR-Curvature (n=1)	0.19	0.37	0.44
Simple and Ordinary Kriging (n=113)	< 0.02	< 0.07	~0.4
Regression Kriging (n=113)	0.06	0.71	0.37

These deficiencies notwithstanding the main problem with the paper is the site specific nature of the analysis, and the lack on any insight into how the results from their study site might be transferred to another site. They present a model that uses surface curvature as the main explanatory variable for soil depth, and then assert that this is the best relationship. However, other than the flawed Kriging analysis they have not demonstrated that other feasible

relationships (e.g. using other explanatory variables found in the literature, for example as listed in the papers above from the DSM community) yield inferior predictors. Thus no case is made that the relationship presented in Equations (1) is anything other than one of many potential feasible predictors of soil depth. More specifically, Kurikose et al. 2009 has surface curvature as one of their ten tested variables, but it is not significant in their regression Kriging (their Table 3). Surface curvature is only the third most significant variable in the analysis of Taylor et al. 2013 (their Figure 4). Shangguan et al. 2016 doesn't find curvature significant at all, though curvature is correlated with MrVBF which was mid-range in their table of explanatory values (their Figure 6).

Response: The reviewer is correct that there have been numerous studies that identify other possible variables that can explain soil thickness. The scope of this project did not include incorporating all of these parameters, largely because this single parameter is relatively easy to obtain from LiDAR-derived DEMs at the appropriate scale and it worked remarkably well at many sites. Perhaps context will also help: we started with soil pit data from our local field site (Johnston Draw). Upon finding a strong relationship between soil thickness and curvature measurements, we expanded our scope to include sites with (1) soil thickness and local curvature measurements reported in the literature and (2) publicly available LiDAR data. We saw similar patterns at most sites despite variations in soil age, climate, latitude, biota, and lithology, and observed that the slope of the TMR-curvature was related to the catchment curvature distribution. Based on this observation, we hypothesized that topographic indices like curvature, at a fine scales, may hold more information that has been previously credited for. At other scales (1, 10, 20, 30, 50 m grid size), other parameters such as LAI or NDVI, as you indicate below, may be superior to curvature, but we're excited to share the observation that at least at this scale, there appears to be a very strong relationship.

All of these studies (including the authors') are regressions between soil depth data and independent variables so do not provide causal relationships.

Response: We encourage and hope that future research may illuminate the exact mechanisms, process, and scale that govern soil thickness; however, this remains a grand challenge in soil science. We believe that this work contributes an important new observation that may prompt further discoveries about the causal mechanisms.

Without some form of causal relationship it is rather speculative to suggest that Equation (1) is universally applicable (in fact their Figure 2A where they examine other independent sites shows considerable scatter using their relationship suggesting that it is not transferable).

Response: We respectfully disagree with the reviewer on the lack of transferability of our model to other sites. Our TMR-curvature approach was created utilizing independent datasets that had comparable sampling protocols (Table 1). In addition, we provide three validation sites of different lithology, climate, and vegetation. We acknowledge the scatter in the predicted data within Gordon Gulch; however, as we stated in the manuscript:

“It is also worth noting that Gordon Gulch also had a larger vertical uncertainty in the LiDAR dataset, 0.175 m compared to 0.034 m for both Babbington and Reynolds Mountain, which may help to explain the lower model performance.” (lines 194)

“In our Idaho study sites (Johnston Draw, Babbington Creek, and Reynolds Mountain), local topography as measured by curvature is the primary determinant of TMR. In contrast, topography explains less of the variation in TMR in catchments with broad curvature distributions. Indeed, uncertainty in the model increases as σ_c increases such that the predictive capability of the model declines in these regions; other physical or biological model parameters may be needed to explain the variability in TMR where surface roughness is high.” (line 224)

Other studies have consistently shown that the strongest independent predictor of soil depth is vegetation. The rationale for this is that vegetation growth is, at least for drier regions, limited by water availability and the deeper the soil the greater is the water storage capacity of the soil. This is not a causal relationship but it is a predictor. An early work that links soil depth and vegetation vigour is Knorr and Lakshmi (2001), and there is now a significant body of research on this to underpin better modelling of latent heat energy exchanges between the land surface and the atmosphere in climate models. Moreover I recall a paper a few years back, but can't seem to find it in my database, that went beyond a simple regression relationship and that coupled remote sensing of vegetation vigour (LAI/NDVI as I recall) with a soil moisture model to estimate soil profile water holding capacity and indirectly soil depth. In principle THIS IS transferable to other catchments, yet vegetation does not rate a mention in the paper being reviewed.

Response: The reviewer is certainly right that there is a lot of research on the relationship between soil depth and vegetation, and possibly an intuitive relationship between soil moisture storage and vegetation characteristics. However, as pointed out by Brantley et al. (2017), there are still many open questions about the interactions between vegetation and soil thickness, not the least of which how to discern between sites in which rooting depth is similar to TMR vs. sites in which rooting depth is much shallower than TMR. We do not attempt to address this complex question in this work, but point out that even without this information, the model appears to work well in many places.

In summary, the novelty of our model is that curvature, a function of hillslope roughness, governs the thickness of the mobile regolith at a resolution between 3 and 5 m, despite variations in lithology, climate, and vegetation, and even where the model performs poorly, it still provides a good first-order estimate of the most abundant soil thickness. We welcome further research that may illuminate the mechanisms that explain why hillslope curvature provides such a great proxy for soil thickness at this scale.

In conclusion while this is an interesting to paper to add to the database of available soil depth data I don't see that it adds significantly to the state-of-the-art, and due to its style of presentation and ignorance of prior art in the DSM community may actually impair progress. I recommend that this paper be rejected.

Response: We disagree that our work would impair the progress of the scientific community. As we have stated in line 31, “To date, soil thickness cannot be efficiently predicted across a landscape. As such, it remains poorly constrained as a key parameter in landscape evolution, hydrological, and earth system models^{12,16,17}.” Our model in fact, makes it possible to acquire such parameters quickly and easily for the first time. In addition, we state in line 53, “...accurate predictions of absolute thicknesses have not been obtained²⁷, and soil thickness models remain over-parameterized and require extensive and computationally

expensive analyses^{24,25,27}.” We would like to highlight that our simple model only needs one soil thickness measurement and high resolution elevation data.

REVIEWERS' COMMENTS:

Reviewer #1 (Remarks to the Author):

Review of Patton et al.

I have now read the revision of "Predicting soil thickness on soil mantled hillslopes". I thank the authors for responding to my queries and I believe they have justified their choices and analysis. I continue to believe this paper to be novel and an important contribution to critical zone science. I imagine figure 2 appearing in many people's talks after publication of this paper. It is fantastic: it shows clear trends, that contradict some of the prevailing theory, yet which make sense under process-based explanations.

I have a few minor comments on the revisions:

I feel like the abstract could do a better job of highlighting the novelty of this contribution. I know the authors are limited by word count. But I just mention a few things that seem like they can be improved:

Lines 12-14: The Heimsath et al papers compared curvature and thickness in 1999 and many papers have done so since then. So at first glance this seems like "a different site gives a different result". But actually here the relationship works across convergent and divergent parts of the landscape and that it applies across lots of places and the regression can be tuned by easily measureable landscape properties. This comes out in the next two sentences but I really think the first three sentences make this paper sound much more incremental than it really is! At a minimum I would include some variation of the phrase "across convergent and divergent parts of the landscape" but also I would just spend some time seeing if the third sentence can be reinforced or integrated with those above. Maybe say something like "We find a linear relationship across diverse landscapes, and also find that the slope of the relationship is correlated with standard deviations of catchment curvature, meaning that despite a wide variety in thicknesses and curvature the relationship can be applied in a wide range of landscapes." Maybe my version is quite clunky and not really better than what is there now, but I would at least work on this component a bit and convince yourselves there is not a clearer way to communicate the novelty.

The sentence on lines 19-21: Again, what is written communicates what is in the paper but I feel that it could highlight the novelty more. Existing methods using kriging need loads of field data that require big field campaigns. The method presented here, which is just as accurate, only requires topographic data. This seems like a big step forward that is only very subtly stated in the abstract.

Line 53: Do you mean have not been obtained by numerical models? I would qualify this sentence.

Line 67-69: In the response to my previous comments the authors defended their selection of the study sites with the assertion that soil pits are the gold standard of soil thickness measurements and they only want to use data collected with pits. I am entirely happy with this rationale.

However, this selection criteria clearly stated here. Maybe say "compared to equivalent datasets"

Line 75: "To allow direct comparison to current literature". If you entered 1+1 into ArcMap and it spit out -200, would you say that you had switched the sign and divided by 100 to "allow direct comparison to current literature"? I would say "ArcMap's curvature function differentiates the slope in percent rather than the actual gradient, and reverses the sign, so to compute curvature values in units 1/m we divide ArcMap output by -100."

REVIEWERS' COMMENTS:

Reviewer #1 (Remarks to the Author):

Review of Patton et al.

I have now read the revision of “Predicting soil thickness on soil mantled hillslopes”. I thank the authors for responding to my queries and I believe they have justified their choices and analysis. I continue to believe this paper to be novel and an important contribution to critical zone science. I imagine figure 2 appearing in many people’s talks after publication of this paper. It is fantastic: it shows clear trends, that contradict some of the prevailing theory, yet which make sense under process-based explanations.

Thank you for this comment and suggestions in the last revision of this manuscript. These revisions really strengthened this manuscript and put the data into better context of the literature and prevailing theory.

I have a few minor comments on the revisions:

I feel like the abstract could do a better job of highlighting the novelty of this contribution. I know the authors are limited by word count. But I just mention a few things that seem like they can be improved: Lines 12-14: The Heimsath et al papers compared curvature and thickness in 1999 and many papers have done so since then. So at first glance this seems like “a different site gives a different result”. But actually here the relationship works across convergent and divergent parts of the landscape and that it applies across lots of places and the regression can be tuned by easily measureable landscape properties. This comes out in the next two sentences but I really think the first three sentences make this paper sound much more incremental than it really is! At a minimum I would include some variation of the phrase “across convergent and divergent parts of the landscape” but also I would just spend some time seeing if the third sentence can be reinforced or integrated with those above. Maybe say something like “We find a linear relationship across diverse landscapes, and also find that the slope of the relationship is correlated with standard deviations of catchment curvature, meaning that despite a wide variety in thicknesses and curvature the relationship can be applied in a wide range of landscapes.” Maybe my version is quite clunky and not really better than what is there now, but I would at least work on this component a bit and convince yourselves there is not a clearer way to communicate the novelty.

The sentence on lines 19-21: Again, what is written communicates what is in the paper but I feel that it could highlight the novelty more. Existing methods using kriging need loads of field data that require big field campaigns. The method presented here, which is just as accurate, only requires topographic data. This seems like a big step forward that is only very subtly stated in the abstract.

Response: Thank you for these suggestions. In response, we revised the abstract to include “across both convergent and divergent parts of the landscape” in Line 12-14. We also revised the rest of the abstract to read “We find similar linear relationships across diverse landscapes (n=6) with the slopes of these relationships varying as a function of the standard deviation in catchment curvatures. This soil thickness-curvature approach is significantly more efficient and just as accurate as kriging-based methods, but requires only high-resolution elevation data and as few as one soil profile.” We also moved “(r²=0.87, RMSE=0.19 m)” to right after “strong linear relationship” to read strong linear relationship (r²=0.87, RMSE=0.19 m)”

We also included “across both convergent and divergent parts of the landscape” in the results section (line 147) to also make this point. We moved in Johnston Draw to the first sentence in the paragraph.

Line 53: Do you mean have not been obtained by numerical models? I would qualify this sentence.

Response: This comment was a bit awkward and hard to interpret but we added “that hinders advancements” to qualify this sentence to read “As such, it remains a poorly constrained yet key parameter that hinders advancements in landscape evolution, hydrological, and earth system models^{12,16,17}.

Line 67-69: In the response to my previous comments the authors defended their selection of the study sites with the assertion that soil pits are the gold standard of soil thickness measurements and they only want to use data collected with pits. I am entirely happy with this rationale. However, this selection criteria clearly stated here. Maybe say “compared to equivalent datasets”

Response: We added “equivalent” and deleted “pre-existing, cross-site TMR-curvature” to simplify sentence.

Line 75: “To allow direct comparison to current literature”. If you entered 1+1 into ArcMap and it spit out -200, would you say that you had switched the sign and divided by 100 to “allow direct comparison to current literature”? I would say “ArcMap’s curvature function differentiates the slope in percent rather than the actual gradient, and reverses the sign, so to compute curvature values in units 1/m we divide ArcMap output by -100.”

Initial comments: I feel like the abstract could do a better job of highlighting the novelty of this contribution.... Line 12 -14 and line 19-21

Response: Thanks for this suggestion. We added “The ArcGIS curvature function differentiates the slope in percent rather than the actual gradient, and reverses the sign, so to compute curvature values in units 1 m^{-1} , we divide the ArcGIS output by -100.”